# Learning Robot Manipulation from Cross-Morphology Demonstration

**Gautam Salhotra**[*] **I-Chun Arthur Liu**[*] **Gaurav S. Sukhatme**[†]
Robotic Embedded Systems Laboratory
University of Southern California
`[salhotra,ichunliu,gaurav]@usc.edu`

**Abstract:** Some Learning from Demonstrations (LfD) methods handle small mismatches in the action spaces of the teacher and student. Here we address the case where the teacher's morphology is substantially different from that of the student. Our framework, Morphological Adaptation in Imitation Learning (**MAIL**), bridges this gap allowing us to train an agent from demonstrations by other agents with significantly different morphologies. **MAIL** learns from suboptimal demonstrations, so long as they provide *some* guidance towards a desired solution. We demonstrate **MAIL** on manipulation tasks with rigid and deformable objects including 3D cloth manipulation interacting with rigid obstacles. We train a visual control policy for a robot with one end-effector using demonstrations from a simulated agent with two end-effectors. **MAIL** shows up to $24\%$ improvement in a normalized performance metric over LfD and non-LfD baselines. It is deployed to a real Franka Panda robot, handles multiple variations in properties for objects (size, rotation, translation), and cloth-specific properties (color, thickness, size, material). An overview is on this [website](website).

**Keywords:** Imitation from Observation, Learning from Demonstration

## 1 Introduction

Learning from Demonstration (LfD) [1, 2] is a set of supervised learning methods where a teacher (often, but not always, a human) demonstrates a task, and a student (usually a robot) uses this information to learn to perform the same task. Some LfD methods cope with small morphological mismatches between the teacher and student [3, 4] (*e.g.,* five-fingered hand to two-fingered gripper). However, they typically fail for a large mismatch (*e.g.,* bimanual human demonstration to a robot arm with one gripper). The key difference is that to reproduce the transition from a demonstration state to the next, no single student action suffices - a sequence of actions may be needed.

Supervised methods are appealing where demonstration-free methods [5] do not converge or underperform [6] and purely analytical approaches are computationally infeasible [7, 8]. In such settings, human demonstrations of complex tasks are often readily available *e.g.,* it is straightforward for a human to show a robot how to fold a cloth. An LfD-based imitation learning approach is appealing in such settings *provided* we allow the human demonstrator to use their body in the way they find most convenient (*e.g.,* using two hands to hang a cloth on a clothesline to dry). This requirement induces a potentially large morphology mismatch - we want to learn and execute complex tasks with deformable objects on a single manipulator robot using natural human demonstrations.

We propose a framework, Morphological Adaptation in Imitation Learning (**MAIL**), to bridge this mismatch. We focus on cases where the number of end-effectors is different from teacher to student,

---

[*]Equal contribution

[†]G.S. Sukhatme holds concurrent appointments as a Professor at USC and as an Amazon Scholar. This paper describes work performed at USC and is not associated with Amazon.

7th Conference on Robot Learning (CoRL 2023), Atlanta, USA.

although the method may be extended to other forms of morphological differences. **MAIL** enables policy learning for a robot with $m$ end-effectors from teachers with $n$ end-effectors. It does not require demonstrator actions, only the states of the objects in the environment making it potentially useful for a variety of end-effectors (pickers, suction gripper, two-fingered grippers, or even hands). It uses trajectory optimization to convert state-based demonstrations into (suboptimal) trajectories in the student's morphology. The optimization uses a learned (forward) dynamics model to trade accuracy for speed, especially useful for tasks with high-dimensional state and observation spaces. The trajectories are then used by an LfD method, optionally with exploration components like reinforcement learning, which is adapted to work with sub-optimal demonstrations and improve upon them by interacting with the environment.

Though the original demonstrations contain states, we generalize the solution to work with image observations in the final policy. We showcase our method on challenging cloth manipulation tasks (Sec. 4.1) for a robot with one end-effector, using image observations, shown in Fig. 1. This setting is challenging for multiple reasons. First, cloth manipulation is easy for bimanual human demonstrators but challenging for a one-handed agent (even humans find cloth manipulation non-trivial with one hand). Second, deformable objects exist in a continuous state space; image observations in this setting are also high-dimensional. Third, the cloth being manipulated makes a large number of con-

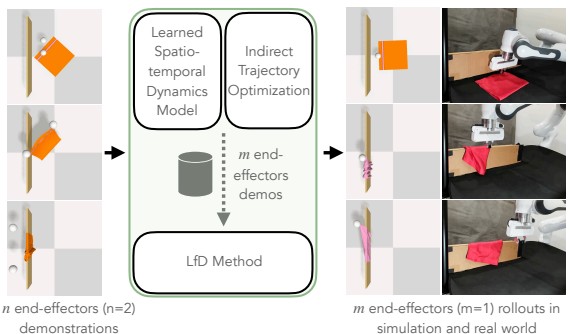

Figure 1: **MAIL** generalizes LfD to large morphological mismatches between teacher and student in difficult manipulation tasks. We show an example task: hang a cloth to dry on a plank (DRY CLOTH). The demonstrations are bimanual, yet the robot learns to execute the task with a single arm and gripper. The learned policy transfers to the real world and is robust to object variations.

tacts (hundreds) that are made/broken per time step. These can *significantly* slow down simulation, and consequently learning and optimization. We make the following contributions:

1. We propose a novel framework, **MAIL**, that bridges the large morphological mismatch in LfD. **MAIL** trains a robot with $m$ end-effectors to learn manipulation from demonstrations with a different ($n$) number of end-effectors.
2. We demonstrate **MAIL** on challenging cloth manipulation tasks on a robot with one end-effector. Our tasks have a high-dimensional ($> 15000$) state space, with several 100 contacts being made/broken per step, and are non-trivial to solve with one end-effector. Our learned agent outperforms baselines by up to 24% on a normalized performance metric and transfers zero-shot to a real robot. We introduce a new variant of 3D cloth manipulation with obstacles - DRY CLOTH.
3. We illustrate **MAIL** providing different instances of end-effector transfer, such as a 3-to-2, 3-to-1, and 2-to-1 end-effector transfer, using a simple rearrangement task with three rigid bodies in simulation and the real world. We further explain how **MAIL** can potentially handle more instances of $n$-to-$m$ end-effector transfer.

## 2 Related Work

**Imitation Learning and Reinforcement Learning with Demonstrations (RLfD):** Imitation learning methods [9, 10, 11, 12, 13] and methods that combine reinforcement learning and demonstrations [14, 15, 1, 2] have shown excellent results in learning a mapping between observations and actions from demonstrations. However, their objective function requires access to the demonstrator's ground truth actions for optimization. This is infeasible for cross-morphology transfer due to action space mismatch. To work around this, prior works have proposed systems for teachers to provide demonstrations in the students' morphology [16] which limits the ability of teachers to efficiently provide data. Similar to imitation learning, offline RL [17, 18, 19] learns from demonstrations stored

in a dataset without online environment interactions. While offline RL can work with large datasets of diverse rollouts to produce generalizable policies [20, 21], it requires the availability of rollouts that have the same action space as the learning agent. **MAIL** learns across morphologies and is not affected by this limitation.

**Imitation from Observation:** Imitation from observation (IfO) methods [3, 9, 22, 23, 24, 25, 26] learn from the states of the demonstration; they do not use state-action pairs. In [27], an approach is proposed to learn repetitive actions using Dynamic Movement Primitives [28] and Bayesian optimization to maximize the similarity between human demonstrations and robot actions. Many IfO methods [3, 23, 24, 29] assume that the student can take a single action to transition from the demonstration's current state to the next state. Some methods [3, 23] use this to train an inverse dynamics model to infer actions. Others extract keypoints from the observations and compute actions by subtracting consecutive keypoint vectors. XIRL [30] uses temporal cycle consistency between demonstrations to learn task progress as a reward function, which is then fed to RL methods. However, when the student has a different action space than the teacher, it may require more than one action for the student to reach consecutive demonstration states. For example, in an object rearrangement task, a two-picker teacher agent can move two objects with one pick-place action. But a one-picker student will need two or more actions to achieve the same result. Zero-shot visual imitation [9] assumes that the statistics of visual observations and agents observations will be similar. However, when solving a task with different numbers of arms, some intermediate states will not be seen in teacher demonstrations. State-of-the-art learning from observation methods [25, 31] have made significant advancements in exploiting information between states. However, their tasks have much longer horizons, hence more states and learning signals than ours. Whether these methods work well on short-horizon, difficult manipulation tasks is uncertain. To address this and provide a meaningful comparison, we conducted experiments to compare **MAIL** with these methods (Sec. 4).

**Trajectory Optimization:** Trajectory optimization algorithms [32, 8, 33] optimize a trajectory by minimizing a cost function, subject to a set of constraints. It has been used for manipulation of rigid and deformable objects [7], even through contact [34] using complementarity constraints [35]. Indirect trajectory optimization only optimizes the actions of a trajectory and uses a simulator for the dynamics instead of adding dynamics constraints at every step.

**Learned Dynamics:** Learning dynamics models is useful when there is no simulator, or if the simulator is too slow or too inaccurate. Learned models have been used with Model-Predictive Control (MPC) to speed up prediction times [36, 37, 38]. A common use case is model-based RL [39], where learning the dynamics is part of the algorithm and has been shown to learn dynamics from states and pixels [40] and applied to real-world tasks [41].

## 3  Formulation and Approach

### 3.1  Preliminaries

We formulate the problem as a partially observable Markov Decision Process (POMDP) with state $s \in \mathcal{S}$, action $a \in \mathcal{A}$, observation $o \in \mathcal{O}$, transition function $\mathcal{T} : \mathcal{S} \times \mathcal{A} \to \mathcal{S}$, horizon $H$, discount factor $\gamma$ and reward function $r : \mathcal{S} \times \mathcal{A} \to \mathbb{R}$. The discounted return at time $t$ is $R_t = \sum_{i=t}^{H} \gamma^i r(s_i, a_i)$ and $s_i \sim \mathcal{T}(s_{i-1}, a_{i-1})$. A task is instantiated with a variant sampled from the task distribution, $v \sim \mathcal{V}$. The initial environment state depends on the task variant, $s_0(v), v \sim \mathcal{V}$. We train a policy $\pi_\theta$ to maximize expected reward $J(\pi_\theta)$ of an episode over task variants $v$, $J(\pi_\theta) = \mathbb{E}_{v \sim \mathcal{V}}[R_0]$, subject to initial state $s_0(v)$ and the dynamics from $\mathcal{T}$.

For an agent with morphology $M$, we differentiate between datasets available as demonstrations ($\mathcal{D}_{Demo}^M$) and those that are optimized ($\mathcal{D}_{Optim}^M$). For our cloth environments, our teacher morphology is two-pickers ($M = 2p$) and student morphology is one-picker ($M = 1p$). We assume the demonstrations are from teachers with a morphology that can be different from the student (and from each other). We refers to these as *teacher* demonstrations, $\mathcal{D}_{Teacher}$, to emphasize that they do not necessarily come from an expert or an oracle. Further, these can be suboptimal. The demonstrations

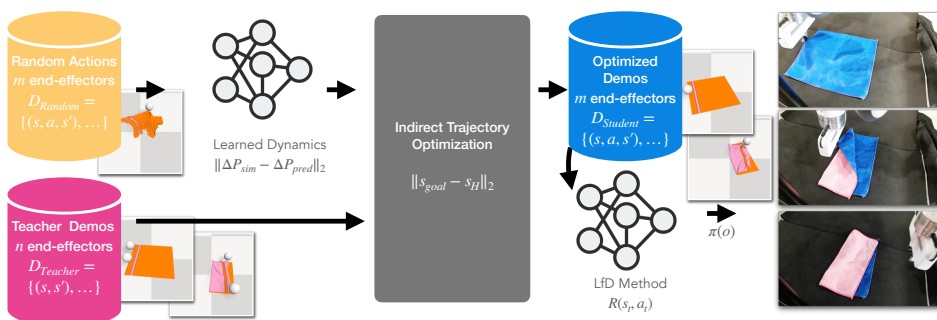

Figure 2: An example cloth folding task with demonstrations from a teacher with $n = 2$ end-effectors, deployed on a Franka Panda with $m = 1$ end-effector (parallel-jaw gripper). We train a network to predict the forward dynamics of the object being manipulated in simulation, using a random action dataset $\mathcal{D}_{Random}$. For every state transition, we match the predicted particle displacements from our model, $\Delta P_{pred}$, to that of the simulator, $\Delta P_{sim}$. Given this learned dynamics and the teacher demonstrations we use indirect trajectory optimization to find student actions that solve the task. The optimization objective is to match with the object states in the demonstration. Finally, we pass the optimized dataset $\mathcal{D}_{Student}$ to a downstream LfD method to get a final policy $\pi$ that generalizes to task variations and extends task learning to image space, enabling real-world deployment.

are state trajectories $\tau_T = (\boldsymbol{s}_0, \ldots, \boldsymbol{s}_{H-1})$. The teacher dataset is made up of $K_T$ such trajectories, $\mathcal{D}_{Teacher} = \{\tau_{T,i}\} \forall i = 1, \ldots, K_T$, using a few task variations from the task distribution $\boldsymbol{v}_d \sim \mathcal{V}$.

We now discuss the components of **MAIL**, shown in Fig. 2. The user provides teacher demonstrations $\mathcal{D}_{Teacher}$. First, we create a dataset of random actions, $\mathcal{D}_{Random}$, and use it to train a dynamics model, $\mathcal{T}_\psi$. $\mathcal{T}_\psi$ reduces computational cost when dealing with contact-rich simulations like cloth manipulation (Sec. 4.1). Next, we convert each teacher demonstration to a trajectory suitable for the student's morphology. For our tasks, we find gradient-free indirect trajectory optimization [33] performs the best (Appendix Sec. A.1). We used $\mathcal{T}_\psi$ for this optimization as it provides the appropriate speed-accuracy trade-off. The optimization objective is to match with object states in the demonstration (we cannot match demonstration actions across morphologies). We combine these optimized trajectories to create a dataset $\mathcal{D}_{Student}$ for the student. Finally, we pass $\mathcal{D}_{Student}$ to a downstream LfD method to learn a policy $\pi$ that generalizes from the task variations in $\mathcal{D}_{Teacher}$ to the task distribution $\mathcal{V}$. It also extends $\pi$ to use image observations and deploys zero-shot on a real robot (rollouts in Fig. 5).

### 3.2 Learned Spatio-temporal Dynamics Model

**MAIL** uses trajectory optimization to convert demonstrations into (suboptimal) trajectories in the student's morphology. This can be prohibitively slow for large state spaces and complex tasks such as cloth manipulation. Robotic simulators have come a long way in advancing fidelity and speed, but simulating complex deformable objects and contact-rich manipulation still requires significant computation making optimization intractable for challenging simulations. We use the NVIDIA FLeX simulator that is based on extended position-based dynamics [42]. We learn a CNN-LSTM based spatio-temporal forward dynamics model with parameters $\psi$, $\mathcal{T}_\psi$, to approximate cloth dynamics, $\mathcal{T}$. This offers a speed-accuracy trade-off with a tractable computation time in environments with large state spaces and complex dynamics. The states of objects are represented as $N$ particle positions: $\boldsymbol{s} = P = \{p_i\}_{i=1\ldots N}$. Each particle state consists of its x, y, and z coordinates. For each task, we generate a corpus of random pick-and-place actions and store them in the dataset $\mathcal{D}_{Random} = \{d_i\}$, where $i = 1, \ldots, K_R$ and $d_i = (P_i, a_i, P_i')$. For each datum $i$, we feed $P_i$ to the CNN network to extract features of particle connectivity. These features are concatenated with $a_i$ and input to the LSTM model to extract features based on the previous particle positions. A fully connected layer followed by layer normalization and $tanh$ activation is used to learn the non-linear combinations of features. The outputs are the predicted particle displacements. The objective function is the

distance between predicted and ground-truth particle displacements, $\|\Delta P_{sim} - \Delta P_{pred}\|_2$. Here $\Delta P_{sim} = \{\Delta p_i\}_{i=1,\dots,N}$ is obtained from the simulator and $\Delta p_i = p_{i+1} - p_i$ for every particle $i$.

Due to its simplicity, the CNN-LSTM dynamics model provides fast inference, compared to a simulator which may have to perform many collision checks at any time step. This speedup is crucial when optimizing over a large state space, as long as the errors in particle positions are tolerable. In our experiments, we were able to get 162 fps with $\mathcal{T}_\psi$, compared to 3.4 fps with the FleX simulator, a 50x speed up (Fig. 8). However, this stage is optional if the environment is low-dimensional, or if the simulation speed-up from inference is not significant. Simulation accuracy is important when training a final policy, to provide *accurate* pick-place locations for execution on a real robot. Hence, the learned dynamics model is not used for training in the downstream LfD method.

## 3.3 Indirect Trajectory Optimization with Learned Dynamics

We use indirect trajectory optimization [33] to find the open-loop action trajectory to match the teacher state trajectory, $\tau_T$. This optimizes for the student's actions while propagating the state with a simulator. We use the learned dynamics $\mathcal{T}_\psi$ to give us fast, approximate optimized trajectories. This is in contrast to direct trajectory optimization (or collocation) that optimizes both states and actions at every time step. Direct trajectory optimization requires dynamics constraints to ensure consistency among states being optimized, which can be challenging for discontinuous dynamics. We use the Cross-Entropy Method (CEM) for optimization, and compare this against other methods, such as SAC (Appendix A.1). Optimization hyperparameters are described in 5. The optimization objective is to match the object's goal state $s_{goal}$ in the demonstration with the same task variant $v_d$. Formally, the problem is defined as:

$$\min_{a_t} \; \|s_{goal} - s_H\|_2 \; \text{ subject to } \; s_0 \; = s_0(v_d) \; \text{ and } \; s_{t+1} = \mathcal{T}(s_t, a_t) \; \forall t = 0, \dots, H-1 \quad (1)$$

where $s_H$ is the predicted final state. Note that if $\tau_T$ has a longer time horizon, it would help to match intermediate states and use multiple-shooting methods. After optimizing the action trajectories for each demonstration $\tau_{T,i} \in \mathcal{D}_{Teacher}$, we use them with the simulator to obtain the optimized trajectories in the student's morphology. These are combined to create the student dataset, $\mathcal{D}_{Student} = \{\tau_1, \tau_2, \tau_3, \dots\}$, where $\tau_i = (s_t, o_t, a_t, s_{t+1}, o_{t+1}, r_t, d) \forall t = 1 \dots H-1$. For generalizability and real-world capabilities, we train an LfD method using $\mathcal{D}_{Student}$. Note that we use the learned dynamics model at this stage, trading faster simulation speed for lower accuracy in the learned model. This is also partially responsible for why $\mathcal{D}_{Student}$ contains suboptimal demonstrations. To reduce the effect of learned model errors, once we obtain the optimized actions, we perform a rollout with the true simulator to get the demonstration data.

## 3.4 Learning from the Optimized Dataset

Our chosen LfD method is DMfD [43], an off-policy RLfD actor-critic method that utilized expert demonstrations as well as rollouts from its own exploration. It learns using an advantage-weighted formulation [44] balanced with an exploration component [5]. As mentioned above, we use the simulator instead of the learned dynamics model $\mathcal{T}_\psi$ at this stage, because accuracy is important in the final reactive policy. Hence, we cannot take the speed-accuracy tradeoff that $\mathcal{T}_\psi$ provides. However, one may choose to use other LfD methods that do not need to interact with the environment [45], in which case neither a simulator nor learned dynamics are needed.

As part of tuning, we employ 100 demonstrations, about two orders of magnitude fewer than the 8000 recommended by the original work. To prevent the policy from overfitting to suboptimal demonstrations in $\mathcal{D}_{Student}$, we disable demonstration-state matching, *i.e.,* resetting the agent to demonstration states and applying imitation reward (see Appendix A.5). These were originally proposed [46] as reference state initialization (RSI). These modifications are essential for our LfD implementation, where the demonstrations do not come from an expert.

From DMfD, the policy $\pi$ is parameterized by parameters $\theta$, and learns from data collected in a replay buffer $\mathcal{B}$. The policy loss contains an advantage-weighted loss $\mathcal{L}_A$ where actions are weighted

by the advantage function $A^\pi(\boldsymbol{s}, \boldsymbol{a}) = Q^\pi(\boldsymbol{s}, \boldsymbol{a}) - V^\pi(\boldsymbol{s})$ and temperature parameter $\lambda$. It also contains an entropy component $\mathcal{L}_E$ to promote exploration during data collection. The final policy loss $\mathcal{L}_\pi$ is a combination of these terms (Eq. 2).

$$\mathcal{L}_A = \underset{\boldsymbol{s},\boldsymbol{a},\boldsymbol{o}\sim\mathcal{B}}{\mathbb{E}}\left[\log\pi_\theta(\boldsymbol{a}|\boldsymbol{o})\exp\left(\frac{1}{\lambda}A^\pi(\boldsymbol{s},\boldsymbol{a})\right)\right] \quad \mathcal{L}_E = \underset{\boldsymbol{s},\boldsymbol{a},\boldsymbol{o}\sim\mathcal{B}}{\mathbb{E}}[\alpha\log\pi_\theta(\boldsymbol{a}|\boldsymbol{o}) - Q(\boldsymbol{s},\boldsymbol{a})]$$

$$\mathcal{L}_\pi = (1-w_E)\mathcal{L}_A + w_E\mathcal{L}_E,\ 0 \le w_E \le 1 \tag{2}$$

where $w_E$ is a tuneable hyper-parameter. The resulting policy is denoted as $\pi_\theta$. We pre-populate buffer $\mathcal{B}$ with $\mathcal{D}_{Student}$. Using LfD, we extend from state inputs to image observations, and generalize from $\boldsymbol{v}_d$ to any variation sampled from $\mathcal{V}$.

## 4 Experiments

Our experiments are designed to answer the following: (1) How does **MAIL** compare to state-of-the-art (SOTA) methods? (Sec. 4.2) (2) How well can **MAIL** solve tasks in the real world? (Sec. 4.2.1) (3) Does **MAIL** generalize to different $n$-to-$m$ end-effector transfers? (Sec. 4.3) Additional experiments demonstrating how different **MAIL** components affect performance are in Appendix A.

### 4.1 Tasks

We experiment with cloth manipulation tasks that are easy for humans to demonstrate but difficult to perform on a robot. We also discuss a simpler rearrangement task with rigid bodies to illustrate generalizability. The tasks are shown in Appendix Fig. 6. We choose a 6-dimensional pick-and-place action space, with xyz positions for pick and place. The end-effectors are pickers in simulation, and a two-finger parallel jaw gripper on the real robot.

CLOTH FOLD: Fold a square cloth in half, along a specified line. DRY CLOTH: Pick up a square cloth from the ground and hang it on a plank to dry, variant of [47]. THREE BOXES: A simple environment with three boxes along a line that needs to be rearranged to designated goal locations. For details on metrics and task variants, see Appendix B.

We use particle positions as the state for training dynamics models and trajectory optimization. We use a 32x32 RGB image as the visual observation, where applicable. We record pre-programmed demonstrations for the teacher dataset for each task. Details of the datasets to train the LfD method and the dynamics model are in Appendix E and Appendix F. The instantaneous reward, used in learning the policy, is the task performance metric at a given state. Further details on architecture and training are in the supplementary material. In all experiments, we compare each method's normalized performance, measured at the end of the task given by $\hat{p}(t) = \frac{p(\boldsymbol{s}_t)-p(\boldsymbol{s}_0)}{p_{opt}-p(\boldsymbol{s}_0)}$, where $p$ is the performance metric of state $\boldsymbol{s}_t$ at time $t$, and $p_{opt}$ is the best performance achievable by the task. We use $\hat{p}(H)$ at the end of the episode ($t = H$).

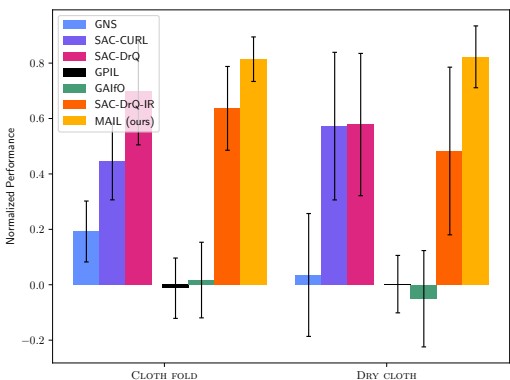

Figure 3: **SOTA performance comparisons.** For each training run, we used the best model in each seed's training run, and evaluated using 100 rollouts across 5 seeds, different from the training seed. Bar height denotes the mean, error bars indicate the standard deviation. **MAIL** outperforms all baselines, in some cases by as much as 24%.

### 4.2 SOTA comparisons

Many LfD baselines (Sec. 2) are not directly applicable, as they do not handle large differences in action space due to different morphologies. We compare **MAIL** with those LfD baselines that produce a policy with image observations, given demonstrations without actions.

1. SAC-CURL [48]: An image-based RL algorithm that uses contrastive learning and SAC [5] as the underlying RL algorithm. It does not require demonstrations.

2. SAC-DrQ [49]: An image-based RL algorithm that uses a regularized Q-function, data augmentations, and SAC as the underlying RL algorithm. It does not require demonstrations.

3. GNS [50]: A SOTA method that represents cloth as a graph and predicts dynamics using a graph neural network (GNN). It does not require demonstrations but learns dynamics from the random actions dataset.We run this learned model with a planner [51], given full state information.

4. SAC-DrQ-IR: A custom variant of SAC [5] that uses DrQ-based [49] image encoding and a state-only imitation reward (IR) to reach the desired state of the object to be manipulated. It does not imitate actions, as they are unavailable.

5. GAIfO [25]: An adversarial imitation learning algorithm that trains a discriminator on state-state pairs $(s, s')$ from both the demonstrator and agent. This is a popular extension of GAIL [13] that learns the same from state-action pairs $(s, a)$.

6. GPIL [31] A goal-directed LfD method that uses demonstrations and agent interactions to learn a goal proximity function. This function provides a dense reward to train a policy.

Fig. 3 has performance comparisons against all baselines. In each environment, the first three columns are demonstration-free baselines, and the last four are LfD methods. **MAIL** outperforms all baselines, in some cases by as much as $24\%$. For the easier CLOTH FOLD task, the SAC-DrQ baseline came within $11\%$ of **MAIL**.

However, all baselines do not perform well in the more difficult DRY CLOTH task. RL methods fail because they have not explored the parameter space enough without guidance from demonstrations. Our custom LfD baseline, SAC-DrQ-IR, has reasonable performance, but the results show that naive imitation alone is not a good form of guidance to solve it. The other LfD baselines, GAIfO and GPIL, have poor performance in both environments. The primary reason is the effect of cross-morphological demonstrations. They perform significantly better with student morphology demonstrations, even if they are suboptimal. Moreover, environment difficulty also plays an important part in the final performance. These and other ablations are in Appendix A.

Surprisingly, the GNS baseline with structured dynamics does not perform well, even though it has been used for cloth modeling [52]. This is

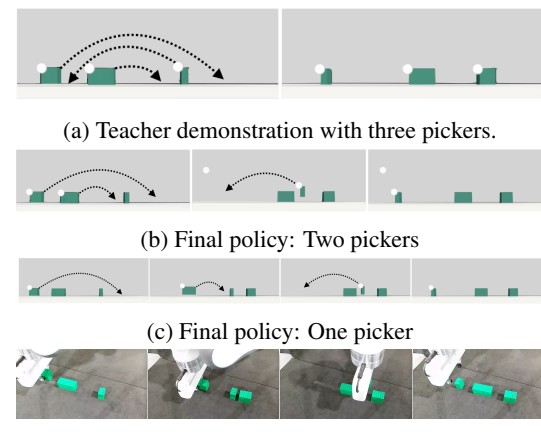

(a) Teacher demonstration with three pickers.

(b) Final policy: Two pickers

(c) Final policy: One picker

(d) Final policy: One Franka Panda robot

Figure 4: **Sample trajectories of the** THREE BOXES **task.** A three-picker teacher trajectory to reach the goal state (Fig. 4a). Final policies of the two-picker and one-picker agent, and real-world execution of the one-picker agent.

because it is designed to learn particle dynamics via small displacements, but our pick-and-place action space enables large displacements. Similar to [51], we break down each pick-and-place action into 100 delta actions to work with the small displacements that GNS is trained on. Thus, planning will accumulate errors from the 100 GNS steps for every action of the planner, which can grow superlinearly due to compounding errors. This makes it difficult to solve the task. It is especially seen in DRY CLOTH, where the displacements required to move the entire cloth over the plank are much higher than the displacements needed for CLOTH FOLD. The rollouts of **MAIL** on DRY CLOTH show the agent following the demonstrated guidance - it learned to hang the cloth over the plank. It also displayed an emergent behavior to straighten out the cloth on the plank to spread it out and achieve higher performance. This was not seen in the two-picker teacher demonstrations.

### 4.2.1 Real-world results

For DRY CLOTH and CLOTH FOLD tasks, we deploy the learned policies on a Franka Panda robot with a single parallel-jaw gripper (Fig. 5, statistics over 10 rollouts). We test the policies with many different variations of square cloth (size, rotation, translation, thickness, color, and material). See Appendix D for performance metrics. The policies achieve $\sim 80\%$ performance, close to the average simulation performance, for both tasks.

### 4.3 Generalizability

We show examples of how **MAIL** learns from a demonstrator with a different number of end-effectors, in a simple THREE BOXES task (Fig. 4). Consider a three-picker agent that solves the task in one pick-place action. Given teacher demonstrations $\mathcal{D}_{Teacher}$, we transfer them into one-picker or two-picker demonstrations using indirect trajectory optimization and the learned dynamics model. These are the optimized datasets that are fed to a downstream LfD method. In both cases, the LfD method learns a model, specific to that morphology, to solve the task. It generalizes from state inputs in the demonstrations to the image inputs received from the environment. Fig. 4 shows the three picker demonstration, a 3-to-2 and 3-to-1 end-effector transfer. We have also done this for the 2-to-1 case (omitted here for brevity). These examples illustrate $n$-to-$m$ end-effector transfer with $n > m$; it is trivial to perform the transfer for $n$-to-$m$ with $n \leq m$ by simply appending the teacher's action space with $m - n$ arms that do no operations.

### 4.4 Limitations

**MAIL** requires object states in demonstrations and during simulation training, however, full state information is not needed at deployment time. It has been tested on the pick-place action space. It has been tested only on cases where the number of end-effectors is different from teacher to student. While it works for high-frequency actions (Appendix A.7), it will likely be difficult to optimize actions to create the student dataset for high-dimensional actions. This is because the curse of dimensionality will apply for larger action spaces when optimizing for $\mathcal{D}_{student}$. The state-visitation distribution of demonstration trajectories must overlap with that of the student agent; this overlap must contain the equilibrium states of the demonstration. For example, a one-gripper agent cannot reach a demonstration state where two objects are moving simultaneously, but it *can* reach a state where both objects are stable at their goal locations (equilibrium). **MAIL** cannot work when the student robot is unable to reach the goal or intermediate states in the demonstration. For example, in trying to open a flimsy bag with two handles, both end-effectors may simultaneously be needed to keep the bag open. When we discuss generalizability for the case $n \leq m$, our chosen method to tackle morphological mismatch is to use fewer arms on the student robot, in lieu of trajectory optimization. This is an inefficient approach since we ignore some arms of the student robot. **MAIL** builds a separate policy for each student robot morphology and each task. While it is possible to train a multi-task policy conditioned on a given task (provided as an embedding or a natural language instruction), extending MAIL to output policies for a variable number of end-effectors would require more careful consideration. Subsequent work could learn a single policy conditioned on the desired morphology - another way to think about a base model for generalized LfD.

## 5  Conclusion

We presented **MAIL**, a framework that enables LfD across morphologies. Our framework enables learning from demonstrations where the number of end-effectors is different from teacher to student. This enables teachers to record demonstrations in the setting of their own morphology, and vastly expands the set of demonstrations to learn from. We show an improvement of up to $24\%$ over SOTA baselines and discuss other baselines that are unable to handle a large mismatch between teacher and student. Our experiments are on challenging household cloth ma-

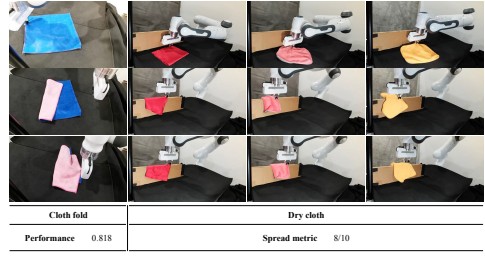

Figure 5: **Real-world results for** CLOTH FOLD **and** DRY CLOTH**.**

nipulation tasks performed by a robot with one end-effector based on bimanual demonstrations. We showed that our policy can be deployed zero-shot on a real Franka Panda robot, and generalizes across cloths of varying size, color, material, thickness, and robustness to cloth rotation and translation. We further showed examples of LfD generalizability with instances of transfer from $n$-to-$m$ end-effectors, with multiple rigid objects. We believe that this is an important step towards allowing LfD to train a robot to learn from *any* robot demonstrations, regardless of robot morphology, expert knowledge, or the medium of demonstration.

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

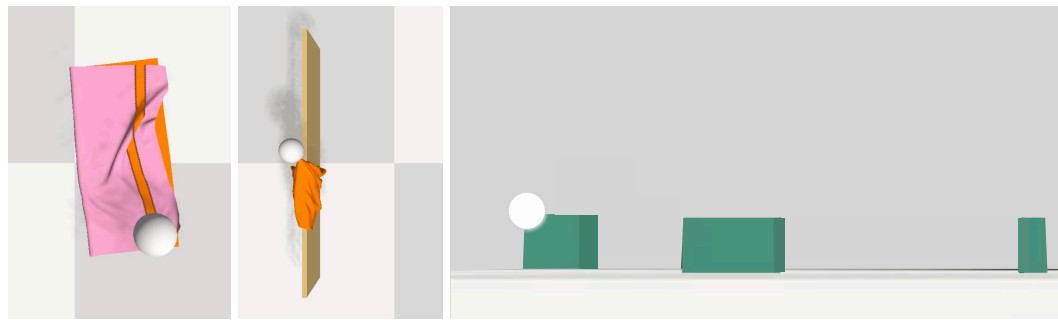

Figure 6: **Environments** used in our experiments, with one end-effector. The end-effectors are pickers (white spheres). In CLOTH FOLD (left) the robot has to fold the cloth (orange and pink) along an edge (inspired by the SoftGym [54] two-picker cloth fold task). In DRY CLOTH (middle) the robot has to hang the cloth (orange and pink) on the drying rack (brown plank). In THREE BOXES (right), the robot has to rearrange three rigid boxes along a line.

## A    Ablations

We use the DRY CLOTH task for all ablations unless specified; it is the most challenging of our tasks. We provide detailed answers to the following questions in Appendix A. Appendix Fig. 7 illustrates the ablations corresponding to each part of the overall method. (1) How do different methods perform in creating optimized dataset $\mathcal{D}_{Student}$? (2) What is the best architecture to learn the task dynamics? (3) How good is $\mathcal{D}_{Student}$ compared to the recorded demonstrations? (4) How well does the downstream LfD method handle different kinds of demonstrations? (5) How does the use of expert state matching affect the downstream LfD? (6) How do the baselines perform across related morphologies and environment?

We discovered that the Cross-Entropy Method (CEM) is the most effective optimizer for generating a $\mathcal{D}_{Student}$ from demonstrations. When combined with CEM, the 1D CNN-LSTM architecture produces the best results for trajectory optimization. Our optimized $\mathcal{D}_{Student}$ performs similarly to the pre-programmed $\mathcal{D}_{Demo}^{1p}$, which has access to full state information of the environment. By utilizing our chosen downstream LfD method, we can successfully complete tasks with a variety of demonstrations and achieve superior performance compared to both $\mathcal{D}_{Student}$ and $\mathcal{D}_{Teacher}$. Expert state matching negatively impacts the performance of DMfD. Lastly, we found that GAIfO trained on our $\mathcal{D}_{Student}$ outperforms GAIfO trained on the $\mathcal{D}_{Teacher}$, and the difficulty of the environment significantly influences the performance of GAIfO and GPIL.

### A.1    Ablate the method for creating optimized dataset $\mathcal{D}_{Student}$

We answer the question: how do different methods perform in creating optimized dataset $\mathcal{D}_{Student}$? We ablate the optimizer used to create $\mathcal{D}_{Student}$ from the demonstrations, labeled ABL1 in Fig. 7, and compare the following methods, given state inputs from $\mathcal{D}_{Teacher}$.

- Random: A trivial random guesser, that serves as a lower benchmark.
- SAC: An RL algorithm that tries to reach the goal states of the demonstrations.
- Covariant Matrix Adaption Evolution Strategy (CMA-ES): An evolutionary strategy that samples optimization parameters from a multi-variate Gaussian, and updates the mean and covariance at each iteration.
- Model Predictive Path Integral (MPPI): An information theoretic MPC algorithm that can support learned dynamics and complex cost criteria [36, 53].
- Cross-Entropy Method (CEM, ours): A well-known gradient-free optimizer, where we assume a Gaussian distribution for optimization parameters.

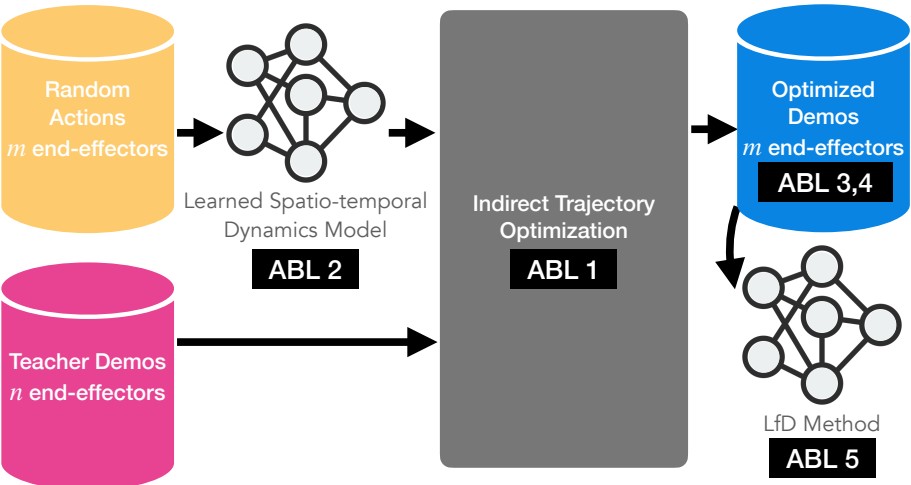

Figure 7: **Ablations** to **MAIL** components.

We did not use gradient-based trajectory optimizers since the contact-rich simulation will give rise to discontinuous dynamics and noisy gradients. As shown in Table 1a, SAC is unable to improve upon the random baseline, likely because of the very large state-space of our environment ($> 15000$ states for $> 5000$ cloth particles) and error accumulations from the imprecision of learned dynamics model. Trajectory optimizers achieve the highest performance, and we chose CEM as the best optimizer based on the performance of the optimized trajectory.

## A.2 Ablate the dynamics model

We answer the question: what is the best architecture to learn the task dynamics? We ablate the learned dynamics model $\mathcal{T}_\psi$, labeled ABL2 in Fig. 7. The environment state is the state from $\mathcal{D}_{Teacher}$ *i.e.,* positions of cloth particles. This is a structured but large state space since the cloth is discretized into $> 5000$ particles.

Table 1b shows the performance of trajectories achieved by using the dynamics models. We see that CNN-LSTM models work better than models that contain only CNNs, graph networks (GNS [50]), transformers (Perceiver IO [55]), or LSTMs. We hypothesize that this is the case since we need to capture the spatial structure of cloth and capture a temporal element across the whole trajectory since particle velocity is not captured in the state. Further, a 1D CNN works better because the cloth state can be simply represented as a 2D vector ($N \times 3$ which represents the xyz for $N$ particles). This is easier to learn with than the 3D state vector fed into 2D CNNs.

GNS performs poorly also due to the reasons of error accumulation from large displacements, discussed in Sec. 4.2. Although Perceiver IO did not perform as well as CNN-LSTM, it did not affect the downstream performance for the LfD method. We conducted an experiment to compare DMfD performance when it was trained on the $\mathcal{D}_{Student}$ obtained from Perceiver IO and CNN-LSTM and found that they had comparable results, shown in Fig. 9. This indicates that **MAIL** is adaptable to different $\mathcal{D}_{Student}$ and capable of learning from suboptimal demonstrations.

Our learned dynamics model $\mathcal{T}_\psi$ was significantly faster than the simulator. We tested it on a simple training run of SAC [5], without parallelization. Our learned dynamics gave 162 fps, about $50x$ faster than the 3.4 fps with the simulator. However, the dynamics error was not insignificant. We compute the state changes in cloth by considering the cloth particles as a point cloud, and computing distances between point clouds using the chamfer distance. We then executed actions on the cloth for the DRY CLOTH task, comparing the cloth state before an action with the model's predicted state and the simulator's true state after the action. Over 100 state transitions, we observed a cloth movement of 0.67 m in the true simulator, and an error of 0.17 m between the true and predicted

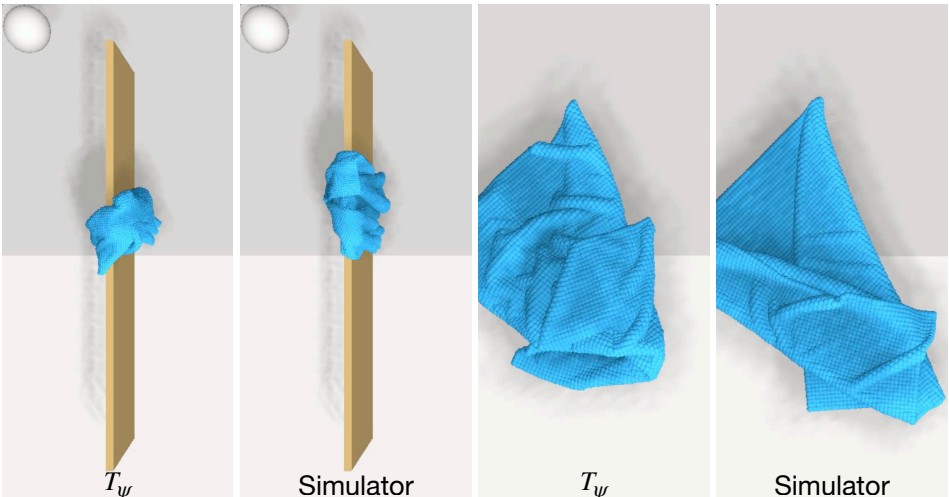

| $\mathcal{T}_\psi$ | Simulator | $\mathcal{T}_\psi$ | Simulator |

Figure 8: **Predictions of the learned spatio-temporal dynamics model $\mathcal{T}_\psi$ and the FleX simulator.** Predictions are made for the same state and action, shown for both cloth tasks. The learned model supports optimization approximately $50x$ faster than the simulator, albeit at the cost of accuracy.

state of the cloth. This accuracy was tolerable for trajectory optimization, qualitatively shown in Fig. 8, where we did not need optimal demonstrations.

### A.3 Compare performance of optimized dataset $\mathcal{D}^{1p}_{Optim}$

We answer the question: how good is $\mathcal{D}_{Student}$ compared to the recorded demonstrations? This ablation gauges the performance of the optimized dataset that we used as the student dataset for LfD, $\mathcal{D}_{Student} = \mathcal{D}^{1p}_{Optim}$. We compare this to other relevant datasets to solve the task, as shown in Table 1c. It is labeled ABL3 in Fig. 7. The two-picker demonstrations $\mathcal{D}^{2p}_{Demo}$ are recorded for an agent with two pickers as end-effectors. This is used as the teacher demonstrations in our experiment $\mathcal{D}_{Teacher} = \mathcal{D}^{2p}_{Demo}$. The one-picker demonstrations $\mathcal{D}^{1p}_{Demo}$ are recorded for an agent with one picker as an end-effector. This is to contrast against the optimized demonstrations in the same morphology, $\mathcal{D}^{1p}_{Optim}$. The random action trajectories are with a one-picker agent, added as a lower performance benchmark. They are the same random trajectories used to train the spatio-temporal dynamics model $\mathcal{T}_\psi$. Naturally, the teacher dataset is the best, as it is trivial to do this task with two pickers. The one-picker dataset has about the same performance as the

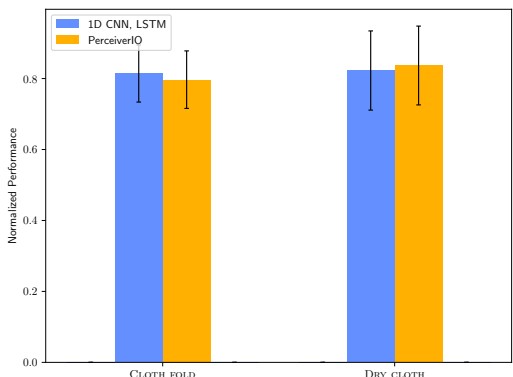

Figure 9: Performance comparison between DMfD trained on $\mathcal{D}_{Student}$ obtained using different learned dynamics models: 1D CNN-LSTM and Perceiver IO. For each training run, we used the best model in each seed's training run, and evaluated using 100 rollouts across 5 seeds, different from the training seed. Bar height denotes the mean, error bars indicate the standard deviation.

optimized dataset $\mathcal{D}^{1p}_{Optim}$, both of which are suboptimal. It can be inferred that it is not trivial to manipulate cloth with one hand. *This is the kind of task we wish to unlock with this work: tasks that are easy to do for teachers in one morphology but difficult to program or record demonstrations for in the student's morphology.* Note that $\mathcal{D}^{1p}_{Optim}$ has been optimized on the fast but inaccurate learned

dynamics model, which is one reason for the reduced performance. This is why the downstream LfD method uses the simulator, as accuracy is very important in the final policy.

| Method | $25^{th}\%$ | $\mu \pm \sigma$ | median | $75^{th}\%$ |
|---|---|---|---|---|
| **Random** | 0.000 | 0.003±0.088 | 0.000 | 0.000 |
| **SAC** | 0.000 | 0.000±0.006 | 0.000 | 0.000 |
| **CMA-ES** | 0.104 | 0.270±0.258 | 0.286 | 0.489 |
| **MPPI** | 0.070 | 0.289±0.264 | 0.275 | 0.474 |
| **CEM** | 0.351 | 0.502±0.242 | 0.501 | 0.702 |

(a) Ablation on the method chosen for creating demonstrations.

| Method | $25^{th}\%$ | $\mu \pm \sigma$ | median | $75^{th}\%$ |
|---|---|---|---|---|
| **Perceiver IO** | 0.305 | 0.450±0.258 | 0.486 | 0.628 |
| **GNS** | -0.182 | 0.002±0.223 | -0.042 | 0.149 |
| **2D CNN, LSTM** | 0.157 | 0.376±0.305 | 0.382 | 0.602 |
| **No CNN, LSTM** | 0.327 | 0.465±0.213 | 0.463 | 0.595 |
| **1D CNN, No LSTM** | 0.202 | 0.407±0.237 | 0.387 | 0.587 |
| **1D CNN, LSTM (ours)** | 0.351 | 0.502±0.242 | 0.501 | 0.702 |

(b) Ablation on the dynamics network architecture.

| Dataset | $25^{th}\%$ | $\mu \pm \sigma$ | median | $75^{th}\%$ |
|---|---|---|---|---|
| $\mathcal{D}_{Random}$ | 0.000 | 0.003±0.088 | 0.000 | 0.000 |
| $\mathcal{D}_{Demo}^{1p}$ | 0.344 | 0.484±0.169 | 0.446 | 0.641 |
| $\mathcal{D}_{Demo}^{2p}$ | 0.696 | 0.744±0.068 | 0.724 | 0.785 |
| $\mathcal{D}_{Optim}^{1p}$ | 0.351 | 0.502±0.242 | 0.501 | 0.702 |

(c) Compare the performance of the optimized dataset.

Table 1: **Ablation results for MAIL**

## A.4   Ablate modality of demonstrations

We answer the question: how well does the downstream LfD method handle different kinds of demonstrations? This ablates the composition of the student dataset fed into LfD, and is labeled ABL4 in Fig. 7. We compare the following datasets for $\mathcal{D}_{Student}$, using the notation for datasets explained in Sec. 3.1:

- Demonstrations in one-picker morphology, $\mathcal{D}_{Demo}^{1p}$: These are non-trivial to create and are thus not as performant, discussed above. Creating these is increasingly difficult as the task becomes more challenging.
- Optimized demos, $\mathcal{D}_{Optim}^{1p}$: This is optimized from the two-picker teacher demonstrations ($\mathcal{D}_{Teacher} = \mathcal{D}_{Demo}^{2p}$), which are easy to collect as the task is trivial with two pickers.
- 50% $\mathcal{D}_{Demo}^{1p}$ and 50% $\mathcal{D}_{Optim}^{1p}$: A mix of trajectories from the two cases above. This is an example of handling multiple demonstrators with different morphologies.

Fig. 10 illustrates that all three variants achieve similar final performance. This demonstrates that the downstream LfD method is capable of solving the task with a variety of suboptimal demonstrations. This could be from one dataset of demonstrations, or even a combination of datasets obtained from a heterogeneous set of teachers.

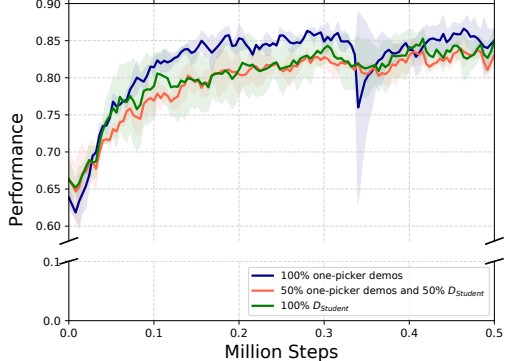
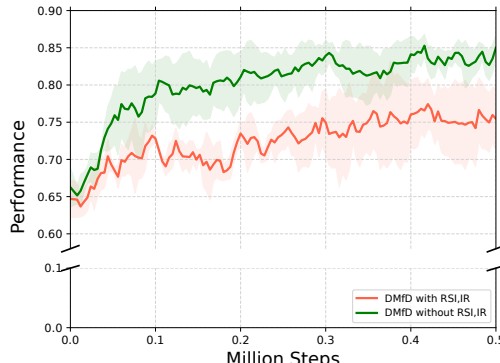

Figure 10: **Ablation on the modality of demonstrations on LfD performance.** Similar performance shows that **MAIL** can learn from a wide variety of demonstrations, or even a mixture of them, without loss in performance. See Sec. A.4.

Figure 11: **Ablation on the effect of reference state initialization (RSI) and imitation reward (IR) on LfD performance.** RSI is not helpful here because our tasks are not as dynamic or long horizon as DeepMimic [46]. See Sec. A.5.

An interesting observation here is that by comparing Fig. 10 and Table 1c, *we see that the final policy is better than the suboptimal demonstrations by a considerable margin, and also slightly improves upon the performance of the teacher demonstrations.* This improvement comes from the LfD method's ability to effectively utilize demonstrations and generalize across task variations. This result, combined with the ablation that we need demonstrations in Sec. 4.2, shows that our downstream LfD method is well adapted to work with suboptimal demonstrations to solve a task.

## A.5 Ablate Reference State Initialization in DMfD

We answer the question: how does the use of demonstration state matching affect the downstream LfD? An improvement we made over the original DMfD algorithm is to disable matching with expert states, known as RSI-IR, first proposed in [46]. We justify this improvement in this ablation, labeled ABL5 in Fig. 7.

As shown in Fig. 11, removing RSI and IR has a net positive effect throughout training, and around 10% on the final policy performance. This means that matching expert states exactly via imitation reward does not help, even during the initial stages of training when the policy is randomly initialized. We believe this is because RSI helps when there are hard-to-reach intermediate states that the policy cannot reach during the initial stages of training. This is true for dynamic or long-horizon tasks, such as karate chops and roundhouse kicks. However, our tasks are quasi-static, and also have a short horizon of 3 for the cloth tasks. In other words, removing this technique allows the policy to freely explore the state space while the demonstrations can still guide the RL policy learning via the advantage-weighted loss from DMfD.

## A.6 Ablate the effect of cross-morphology on LfD baselines

We answer the question: how do established LfD baselines perform across morphologies? This ablation studies the effect of cross-morphology in the demonstrations, where we compare the performance of GAIfO, when provided demonstrations from the teacher dataset $\mathcal{D}_{Teacher}$ and (suboptimal) student dataset $\mathcal{D}_{Student}$, for the DRY CLOTH task.

As we can see in Table 2, there is a 36% performance improvement when using $\mathcal{D}_{Student}$ instead of $\mathcal{D}_{Teacher}$. The primary difference that the agent sees during training is the richness of demonstration states, as the demonstration actions are not available to learn from. Since the student morphology has only one picker, any demonstration for the task (DRY CLOTH) includes multiple intermediate states of the cloth in various conditions of being partially hung for drying. By contrast, the teacher

requires fewer pick-place steps to complete the task, and thus there are fewer intermediate states in the demonstrations.

## A.7 Ablate the effect of environment difficulty on LfD baselines

We answer the question: how do established LfD baselines perform across environments? Given the subpar performance of the LfD baselines GAIfO and GPIL on our SOTA environments, we ablated the effect of environment difficulty. We took the easy cloth environment (CLOTH FOLD) and used an easier variant of it, CLOTH FOLD DIAGONAL PINNED [43]. In this variant, the agent has to fold cloth along a diagonal, which can be done by manipulating only one corner of the cloth. Moreover, one corner of the cloth is pinned to prevent sliding, making it easier to perform. We used state-based observations and a small-displacement action space, where the agent outputs incremental picker displacements instead of pick-and-place locations. We can see in Table 3 that the same baselines are able to perform *significantly* better in this environment. Hence, we believe manipulating with long-horizon pick-place actions, with an image observation, makes it challenging for the baselines to perform cloth manipulation tasks described in Sec. 4.1 and Appendix B.

| Method | $25^{th}\%$ | $\mu \pm \sigma$ | median | $75^{th}\%$ |
|---|---|---|---|---|
| $D_{Teacher}$ | -0.198 | -0.055±0.183 | -0.043 | 0.078 |
| $D_{Student}$ | 0.199 | 0.363±0.245 | 0.409 | 0.528 |

Table 2: Ablation of GAIfO on the effect of cross-morphology. We compare the normalized performance, measured at the end of the task.

| Method | $25^{th}\%$ | $\mu \pm \sigma$ | median | $75^{th}\%$ |
|---|---|---|---|---|
| GPIL | 0.356 | 0.427±0.162 | 0.487 | 0.553 |
| GAIfO | 0.115 | 0.374±0.267 | 0.471 | 0.592 |

Table 3: Measuring performance on the easy cloth task, CLOTH FOLD DIAGONAL PINNED. We compare the normalized performance, measured at the end of the task.

## B Tasks

Here we give more details about the tasks, including the performance functions, teacher dataset, and sample images. Fig. 6 shows images all of simulation environments used for SOTA comparisons and generalizability, with one end-effector. In each environment, the end-effectors are pickers (white spheres). In cloth-based environments, the cloth is discretized into an 80x80 grid of particles, giving a total of 6400 particles.

1. CLOTH FOLD: Fold a square cloth in half, along a specified line. The performance metric is the distance of the cloth particles left of the folding line, to those on the right of the folding line. A fully folded cloth should have these two halves virtually overlap. Teacher demonstrations are from an agent with two pickers (*i.e.,* $\mathcal{D}_{Teacher} = \mathcal{D}_{Demo}^{2p}$); we solve the task on a student agent with one picker. Task variations are in cloth rotation.

2. DRY CLOTH: Pick up a square cloth from the ground and hang it on a plank to dry, variant of [47]. The performance metric is the number of cloth particles (in simulation) on either side of the plank and above the ground. Teacher demonstrations are from an agent with two pickers (*i.e.,* $\mathcal{D}_{Teacher} = \mathcal{D}_{Demo}^{2p}$); we solve the task on a student agent with one picker. Task variations are in cloth rotations and translations with respect to the plank.

3. THREE BOXES: A simple environment with three boxes along a line that need to be rearranged to designated goal locations. Teacher demonstrations are from an agent with three pickers (*i.e.*, $\mathcal{D}_{Teacher} = \mathcal{D}_{Demo}^{3p}$); we solve the task on student agents with one picker and two pickers. Performance is measured by the distance of each object from its goal location. This task is used to illustrate the generalizability of **MAIL** with various $n$-to-$m$ end-effector transfers, and is not used in the SOTA comparisons.

## C  Hyperparameter choices for MAIL

In this section, Table 4 shows the hyperparameters chosen for training the forward dynamics model $\mathcal{T}_\psi$. Table 5 shows the details of CEM hyperparameter choices. Table 6 shows the hyperparameters for our chosen LfD method (DMfD).

| Parameter | Description |
|---|---|
| **CNN** | 4 layers, 32 channels, 3x3 kernel, leaky ReLU activation. stride = 2 for the first layer, stride = 1 for subsequent layers |
| **LSTM** | One layer Hidden size = 32 |
| **Other Parameters** | Learning rate $\alpha =$1e-5 Batch size = 128 |

Table 4: Hyper-parameters for training the forward dynamics model.

| | Planning Horizon | Number of optimization iterations | Number of env interactions |
|---|---|---|---|
| **1** | 1 | 2 | 21,000 |
| **2** | 2 | 2 | 15,000 |
| **3** | 2 | 2 | 21,000 |
| **4** | 2 | 2 | 31,000 |
| **5** | 2 | 2 | 34,000 |
| **6** | 2 | 10 | 21,000 |
| **7** | 2 | 1 | 21,000 |
| **8** | 2 | 1 | 15,000 |
| **9** | 2 | 1 | 32,000 |
| **10** | 3 | 2 | 21,000 |
| **11** | 3 | 10 | 21,000 |
| **12** | 4 | 2 | 21,000 |
| **13** | 4 | 10 | 21,000 |

Table 5: CEM hyper-parameters tested for tuning the trajectory optimization. We conducted ten rollouts for each parameter set and used the set with the highest average normalized performance on the teacher demonstrations. Population size is determined by the number of environment interactions. The number of elites for each CEM iteration is 10% of population size.

## D  Performance metrics for real-world cloth experiments

In this section, we explain the metrics for measuring performance of the cloth, to explain the sim2real results discussed in Sec. 4.2.1

For CLOTH FOLD task, we measure performance at time $t$ by the number of pixels of the top color $pix_{top,t}$ and bottom color $pix_{bot,t}$ of the flattened cloth, compared to the maximum number of pixels, $pix_{max}$ (Fig. 12).

For DRY CLOTH task, it is challenging to measure pixels on the sides and top of the plank. Moreover, we could be double counting pixels if they are visible in both side and top views. Hence, we measure

| Parameter | Description |
|---|---|
| **State encoding** | Fully connected network (FCN) 
 2 hidden layers of 1024, ReLU activation |
| **Image encoding** | 32x32 RGB input, with random crops. 
 CNN: 4 layers, 32 channels, stride 1, 3x3 kernel, leaky ReLU activation 
 FCN: 1 layer of 1024 neurons, $tanh$ activation |
| **Actor** | Fully connected network 
 2 hidden layers of 1024, leaky ReLU activation |
| **Critic** | Fully connected network 
 2 hidden layers of 1024, leaky ReLU activation |
| **Other parameters** | Discount factor: $\gamma = 0.9$ 
 Entropy loss weight: $w_E = 0.1$ 
 Entropy regularizer coefficient: $\alpha = 0.5$ 
 Batch size = 256 
 Replay buffer size = 600,000 
 RSI-IR probability = 0 (disabled) |

Table 6: Hyper-parameters used in the LfD method (DMfD).

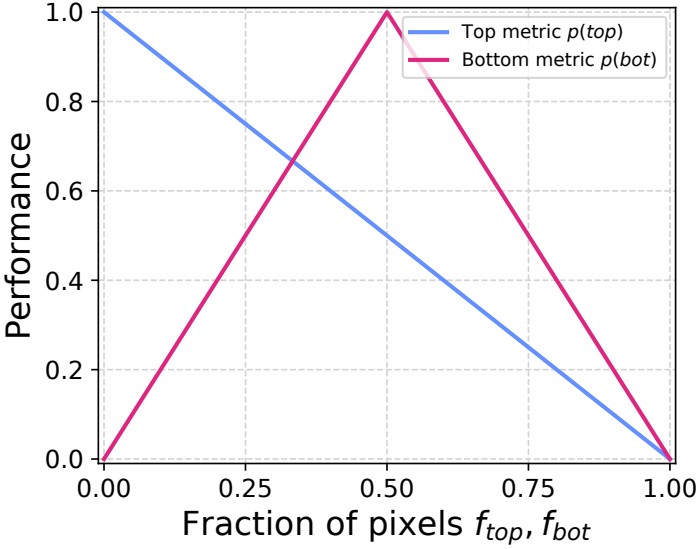

Figure 12: **Performance function for** CLOTH FOLD **on the real robot.** At time $t$, we measure the fraction of pixels visible to the maximum number of pixels visible $f_{top} = pix_{top,t}/pix_{max}$ and $f_{bot} = pix_{bot,t}/pix_{max}$. Performance for the top of the cloth should be 1 when it is not visible, $p(top) = 1 - f_{top}$. Performance for the bottom of the cloth should be 1 when it is exactly half-folded on top of the top side, $p(bot) = \min[2(1 - f_{bot}), 2f_{bot}]$. Final performance is an average of both metrics, $p(s_t) = p(top) + p(bottom)/2$. Note that the cloth is flattened at the start, thus $pix_{max} = pix_{top,0}$.

the cloth to determine whether the length of the cloth *on top of* the plank is equal to or greater than the side of the square cloth. We call this the spread metric.

The policies achieve $\sim 80\%$ performance, which is about the average performance of our method in simulation, for both tasks. However, since these performance metrics are different in the simulation and real world, we cannot *quantify* the sim2real gap through these numbers.

## E    Collected dataset of teacher demonstrations

We have 100 demonstrations provided by the teacher, mentioned on Sec. 3.4. The diversity of the task comes from the initial conditions for these demonstrations, which are sampled from the task distribution $v_d \sim \mathcal{V}$. This variability in the initial state adds diversity to the dataset. The quality and performance of these teacher demonstrations were briefly discussed in the ablations (Sec. A.4).

All demonstrations come from a scripted policy. For ClothFold, the teacher has two end-effectors and picks two corners of the cloth to move them towards the other two corners. For DryCloth, the teacher has two end-effectors and picks two corners of the cloth to move them to the other side of the rack. They maintain the same distance between each other during the move to ensure the cloth is spread out when it hangs on the rack. For ThreeBoxes, the teacher has three end-effectors. It picks up all the boxes simultaneously and places them in their respective goals.

## F    Random actions dataset used for training the dynamics model

We trained the dynamics model on random actions from various states, to cover the state-action distributions our tasks would operate under.

For CLOTH FOLD, our random action policy is to pick a random cloth particle and move the particle to a random goal location within the action space. For DRY CLOTH, the random action policy is to pick a random cloth particle, and move it to a random goal location around the drying rack, to learn cloth interactions around the rack. For completeness, we also trained a forward dynamics model for the THREE BOXES task. Here, the random action policy is to pick the boxes in order and sample a random place location within the action space.

Each task's episode horizon is 3. Our actions are pick-and-place actions, and the action space is in the full range of visibility of the camera. For DRY CLOTH, this limit is $[-0.5, 0, 0.5]$ to $[0.5, 0.7, 0.5]$. For CLOTH FOLD it is $[-0.9, 0, 0.9]$ to $[0.9, 0.7, 0.9]$. For THREE BOXES it is $-0.1$ to $1.35$.

