# OpenReview forum: "Learning Robot Manipulation from Cross-Morphology Demonstration"
_robot-learning.org/CoRL/2023/Conference — CoRL 2023 Poster_

### Official Review · Reviewer_UU6D · 2023-06-25

**Confidence:** 4
**Originality:** Fair
**Technical Quality:** Fair
**Clarity Of Presentation:** Fair
**Impact:** 3

**Recommendation:**

Weak Accept: I recommend accepting the paper, but will not argue for my recommendation if the majority of other reviewers have a different opinion.

**Review:**

Strengths:

1. Learning from cross-morphology demonstrations is an important problem especially when we consider humans providing demonstrations to robots, due to completely different morphologies.

2. While the paper only tests on two variants of cloth manipulation (as well as a less-interesting simulated block arrangement task), it benchmarks against a lot of other methods, so I appreciate that rigorous experimentation. There are multiple variants of SAC.

3. Cloth manipulation is itself challenging and I appreciate their efforts in doing so.

Areas to potentially improve:

I think I get the rough idea of the method but there are a lot of confusing aspects to the paper which hinder my understanding. Here are my confusions:

- While the method, MAIL, is advertised as an imitation learning method, it is really a combination of imitation learning and reinforcement learning, with the RL part happening later. This could make it harder to apply for robotics tasks that don't have a suitable simulator. Maybe the RL aspect should be emphasized earlier just so that the reader is aware that RL will be needed.

- Similarly in terms of clarification, MAIL specifically focuses on *end-effector count* morphology, such as using 1 picker or 2 picker. This is somewhat different from what I had thought of originally which could have referred to any human demonstration versus robot demonstration (as humans and robots have different morphologies).

- There are two places in the text that suggest cloth manipulation is very slow to simulate, in lines 62-63 ("These can significantly slow down simulation…") and in lines 147-152 ("Robotic simulators have come a long way in advancing fidelity and speed, but [...] making optimization intractable for challenging simulations"). It took me a while to understand what the paper was saying, since I had assumed their method would try to *avoid* any cloth simulation due to "intractability" but the paper quickly says that it will use NVIDIA FleX simulation anyway! After reading carefully, I think the point is that MAIL still uses NVIDIA FleX, it just doesn't use it for "forward dynamics", instead it uses a learned dynamics model which imitates the simulator but is faster to query than the simulator itself. I think that's the reason (please correct me if I am wrong) but I fear the writing is confusing in that regard.

- As a side comment, I have used NVIDIA's cloth simulation in the past and I found the simulation to be quite fast to run, hence I'm surprised that the paper reports a 50X speed-up (162 fps vs 3.4 fps) with the learned dynamics vs the underlying simulator. Since the paper uses CEM for optimization, isn't it possible to parallelize the underlying FleX simulation to get faster data (even if each individual environment runs at the same speed)?

- Lines 152-153: the paper says it uses a CNN-LSTM based forward dynamics model. But then why are the inputs represented as N particle positions? Normally, a CNN would take in an image as input, right? Taking in particles would imply using a graph neural network or a point cloud neural network, where I might interpret a "particle" as a graph node or a single point, respectively. Incidentally, why use CNNs and LSTMs at all if one can use Transformers?

- Sections 3.3 and 3.4: I was getting confused about why the learned dynamics model was used in 3.3, yet the underlying simulator was used in 3.4. Two comments: first, maybe it would help to clearly identify when the learned dynamics versus the underlying simulator is used in the different stages of MAIL. Second, since the underlying simulator is used in 3.4 anyway, that seems to suggest the underlying cloth simulation is fast enough? Just another thought: maybe it helps to identify the rough computational resources needed for both 3.3 and 3.4 to understand which of those two stages is the time bottleneck.

Other concerns have to do with some possible over-claiming:

- Line 72 says that "MAIL can handle general instances of $n$-to-$m$ end-effector transfer. However this is not what is being shown. Instead, as line 74 says, the specific example MAIL shows is handling 3-to-2, 3-to-1, and 2-to-1 end-effector transfer. Saying "general instances" implies that we can have $n$ and $m$ be arbitrary integers. Thus I would suggest changing line 72 accordingly.

- Line 318 to 319: similarly, the paper says "Thus, MAIL is capable of general $n$-to-$m$ end-effector transfer. This is not what was shown. Also just before this, line 316, I would change "Thus, MAIL can solve a task using…" to "Thus, MAIL can solve this particular task using…" because this is focusing on the particular task of block rearrangement.

- Line 368 (in the conclusion), similarly, the paper says "We further showed LfD generalizability to any transfer from $n$-to-$m$ end-effectors." I suggest rewriting the sentence to say "3-to-2, 3-to-1, and 2-to-1" because that is what was actually shown in experiments, and it should condition on the specific tasks that were shown.

Finally, one comment has to do with related work. I would recommend adding more material about cross-morphology learning. For example, cross-embodied IRL https://x-irl.github.io/ and AVID https://arxiv.org/abs/1912.04443 which tries to translate demonstrations from humans to robots. These are just two examples I am aware of; there are likely many more especially if we count imitation from dexterous manipulators (e.g,. Human hand vs Allegro hand). I would probably de-emphasize the generic "Imitation from Observation" portion unless those specifically deal with the challenges of translating from human demonstrations to robot demonstrations.


**Quality Of The Limitations Section:**

Limitations are addressed clearly

**Questions For Rebuttal:**

I don't have specific questions other than those implied by the review above. Please correct me if I happen to be mistaken in my understandings.


**Robotics Focus:**

Sufficient demonstration on hardware

**Summary Of Paper:**

This paper studies learning from demonstrations that have substantially different morphologies than the current robot, hence the use of the term "cross-morphology". The method is named "Morphological Adaptation in Imitation Learning (MAIL)" and it tries to tackle policy learning for a robot with $m$ end-effectors from demonstrators with $n$ end-effectors, where in general $n \ne m$. Since the end-effector count is not necessarily equal, the key idea of MAIL is not to match morphologies but to instead match the object states instead. The paper tests on robotic cloth manipulation applications including simulation and in the real world, along with a simulated block arrangement task. They show that their method is able to outperform numerous competing baseline methods.


**Summary Of Recommendation:**

This method and paper has some promise. There are a few concerns I have as noted in the review about how much this will generalize to $m$ and $n$ end-effectors as well as some confusing aspects of the text.

Update: based on the author response where they said they would modify the writing of the paper about the "any n-to-m transfer" part, I'm OK with a weak accept.

---

### Official Review · Reviewer_nE7X · 2023-07-11

**Confidence:** 4
**Originality:** Very Good
**Technical Quality:** Very Good
**Clarity Of Presentation:** Fair
**Impact:** 2

**Recommendation:**

Weak Accept: I recommend accepting the paper, but will not argue for my recommendation if the majority of other reviewers have a different opinion.

**Review:**

The paper demonstrates a high level of quality in terms of its methodology, experimental setup, and results. The authors propose a novel framework, MAIL, and provide a detailed explanation of its components. The experiments are well-designed and thorough, showcasing the effectiveness of the framework on challenging cloth manipulation tasks . The use of both simulated and real-world scenarios adds to the robustness of the study.

Clarity: The paper is generally well-written and effectively conveys the main ideas and contributions of the work. The authors provide clear explanations of the framework and its components, making it accessible to readers. However, some sections could benefit from further clarification, especially regarding specific technical details related to trajectory optimization and the adaptation of LfD methods.

Originality: The paper introduces a novel framework, MAIL, which addresses the problem of learning from demonstrations across robots with different morphologies. While the concept of LfD is not new, the specific focus on morphological adaptation is a novel contribution. The approach of converting state-based demonstrations into suboptimal trajectories and leveraging trajectory optimization for speed and efficiency is also original. The combination of these elements distinguishes the paper from previous works.

Significance: The work presented in the paper has significant implications for the field of imitation learning and robotics. The ability to train robots with different morphologies using demonstrations from other robots expands the possibilities for knowledge transfer and robot learning. The demonstrated improvements over baselines and successful deployment on a real robot highlight the practical relevance of the framework. The generalizability of the approach to different tasks and end-effector transfers further enhances its significance.

Strengths:
1. Introduction of the novel framework MAIL, addressing the problem of morphological adaptation in LfD.
2. Thorough experimental evaluation showcasing the effectiveness of MAIL on challenging manipulation tasks.
3. Successful deployment of the learned policy on a real robot, demonstrating practical applicability.
4. Generalizability of the framework to various instances of end-effector transfer, illustrating its versatility.

Weaknesses:
1. Limited Discussion on Limitations: The paper does not extensively discuss the limitations of the proposed framework. Addressing potential challenges, constraints, or scenarios where the framework might not perform well would provide a more balanced perspective on its applicability. In line 317, where n<=m, the author provided a method to tackle mismatch when the teacher has less action space than the robot. However, such an approach would limit the capability of the robot and seems to be a wasteful approach.
2. Incomplete Technical Explanations: Some technical aspects, such as trajectory optimization and the adaptation of LfD methods, could be further clarified. Improved explanations and additional details would help readers better understand the proposed framework and its underlying mechanisms (as creating the optimized student dataset seems to be a key point in this paper).
3. Limited Exploration of Alternative Optimization Techniques: The paper primarily focuses on trajectory optimization as the means to convert demonstrations. Exploring and discussing alternative optimization techniques or considering the limitations of the chosen approach would enhance the robustness of the framework.
4. Lack of Discussion on Dataset Variability: The paper does not discuss the variability or diversity of the datasets used for training and evaluation. Exploring the impact of dataset characteristics, such as the number of demonstrations, their quality, or the diversity of tasks, would provide deeper insights into the framework's performance.


**Quality Of The Limitations Section:**

Additional details required

**Questions For Rebuttal:**

1. Some technical details related to trajectory optimization and adapting LfD methods could be further clarified. In particular, the sole purpose of the learned dynamics model seems to be only for faster simulation speed for generating the student dataset (and presumably, this would be an offline process). The same model is not used for the actual LfD (line 199). What is the effect of using the learned dynamic as a proxy for generating the student dataset?
2. Moreover, for the metric in Figure 3, are they comparing against the student dataset? If so, wouldn't this be affected by the accuracy of the trajectory optimizer?
3. The typical optimizers for robotics seem to have not been considered. E.g., STOMP, CHOMP, TrajOpt, etc. Some of which are also gradient-free.


Minor comments:
- The notation style in Figure 2 mismatches with the rest of the paper.
- It would be better if the figure could be self-contained. Many of the minimization objectives presented in the figure are unclear (due to undefined notations) unless you jump to their respective sections.
- Are there performance metrics for the Three-Box task?

**Robotics Focus:**

Sufficient demonstration on hardware

**Summary Of Paper:**

The paper introduces a framework called Morphological Adaptation in Imitation Learning (MAIL), enabling learning from demonstrations across robots with different morphologies. It addresses the challenge of training a robot with a different number of end-effectors compared to the teacher robot. Using trajectory optimization, the framework converts state-based demonstrations into suboptimal trajectories in the student's morphology. These trajectories are then used by a Learning from Demonstrations (LfD) method to improve the student's performance by interacting with the environment.

The paper's main contributions are the proposal of the MAIL framework that allows training a robot with demonstrations that has a morphological mismatch between end-effectors. The paper demonstrates MAIL's effectiveness on manipulation tasks involving rigid and deformable objects, including 3D cloth manipulation with obstacles. The policy learned using MAIL also transfers successfully to a real robot.

**Summary Of Recommendation:**

I think the paper demonstrates a novel framework and presents valuable contributions in the field of morphological adaptation in imitation learning.

-----------------------------

**Post rebuttal**

The author had addressed some of my concerns on the contribution and the lack of details in the setup. I think this work does have value in domain adaptation for learning from demonstration.

---

> ### Author Response · Authors · 2023-08-11
> **Response to Reviewer nE7X (1/2)**
>
> We thank you for the insightful comments on our work. We have addressed your concerns and queries in this response.
> Each comment of yours is in bold, followed by our response.
>
> 1.  **Some technical details related to trajectory optimization and adapting LfD methods could be further clarified**:
>     We will add more technical explanations to our paper.
>     Specifically, the final version will include details of how our trajectory optimization is formulated, some of which are already present: the fitness function, dynamics function, CEM hyperparameters, action parameters chosen to be optimized, and obtaining states once the actions are optimized.
>
>     Similarly, we will include these details about the adaptation of LfD methods: the optimized demonstrations $\mathcal{D}_{student}$ we provide, details about the optimized dataset and its characteristics, the chosen LfD method, and some other alternatives (this can include offline alternatives that don't need environment interaction, so we don't need the simulator or the learned dynamics model for them).
>
>     If there is any other relevant detail you think we may have missed, please let us know.
>
>
> 1.  **What is the effect of using the learned dynamic as a proxy for generating the student dataset?**: The learned dynamics model will have *some* error compared to true dynamics, described as the speed-accuracy tradeoff in section 3.2.
>     Thus, the student dataset is optimized based on fast but approximate dynamics, resulting in a fast but approximate calculation of the fitness function.
>     Once we get the optimized action sequence, we pass it through the simulator to get the true optimized states and true performance.
>     This means the dataset performance during CEM will be off from the true performance when measured by passing the actions through the true dynamics.
>     The numbers we state in our paper are those of passing the optimized actions through the true dynamics (FLeX simulator).
>     Another effect of using the learned dynamics is that trajectory optimization is a lot faster because the dynamics is about 50x faster (see line 169 or Figure 8).
>     Note that we cannot use the learned dynamics for the LfD method. We cannot afford the speed-accuracy tradeoff, since we need a good reactive policy that works with true dynamics. This is explained in line 199.
> 1.  **for the metric in Figure 3, are they comparing against the student dataset? If so, wouldn't this be affected by the accuracy of the trajectory optimizer?**:
>     In Figure 3 (SOTA performance comparisons), we are comparing **MAIL** against the baselines. **MAIL** includes one stage that produces $\mathcal{D}_{student}$, but the final numbers in Figure 3 are after we have finished all stages (after the LfD method). Yes, it is affected by both the accuracy and performance of the trajectory optimizer.
>     Because of the importance of the optimization stage, we have added ablations that discussed the effect of
>     the chosen trajectory optimizer (sec A.2.1),
>     different learned dynamics models that would affect the trajectory optimizer (sec A.2.2), and
>     the performance of the trajectory optimization (sec A.2.3).
> 1.  **The typical optimizers for robotics seem to have not been considered. E.g., STOMP, CHOMP, TrajOpt, etc. Some of which are also gradient-free**:
>     The suggested methods mainly do state trajectory optimization as part of kinematic motion planning.
>     Once the states are optimized, it is assumed the actions are obtainable after optimization.
>     However, environments involving cloth manipulation are heavily underactuated. Thus, obtaining the right action $a$ from state $s$ to $s'$ is not trivial.
>     One way to use such planners could be to train an inverse dynamics model to infer actions from an optimized state trajectory. We tried training such an inverse dynamics model earlier in the project and found it prohibitively costly to train a model that takes in 2 inputs of $>15000$ dimensions (states $s$ and $s'$).
>     Further, some state transitions may take multiple actions to go from $s$ to $s'$, making it further unsuitable to train.
>
>     However, Model Predictive Path Integral (MPPI)[1] is a suitable robotics MPC method to compare against.
>     MPPI will directly optimize the actions to reach the goal, given a dynamics model.
>     The experiment is running; we will update you as soon as it finishes.
>
> 1.  **Minor comments**:
>     We will update Fig. 2 with the appropriate notation style, especially for the datasets.
>     With regard to the minimization objectives, we will explain the notation in the caption. We thought about removing the objectives from the figure or explaining them briefly in words but are leaning towards providing an explanation in the caption.
>     In the ThreeBoxes task, our only metric was the appropriate relative positions of the three blocks after rearrangement.
>     We qualitatively observed that it achieved success in 10/10 trials.
>
> (... continued in 2/2 below)

---

> > ### Author Response · Authors · 2023-08-11
> > **Response to Reviewer nE7X (2/2)**
> >
> > (... continued from 1/2 above)
> >
> > Beyond the questions for rebuttal, we also wanted to address other weaknesses mentioned in the review.
> >
> > 1.  **Limited Discussion on Limitations**:
> >     We thank you for this opportunity to explain the limitations more, as we believe it is essential to list both benefits and downsides for a good scientific publication. We have expanded the limitations sections as below and will add it to the paper.
> >
> >     1.  **MAIL** requires object states in demonstrations and during simulation training, however, full state information is not needed at deployment time.
> >     1.  **MAIL** has been tested for environments with a pick-place action space.
> >         While it works for high-frequency actions (Sec. A.2.7), it will likely be difficult to optimize actions to create the student dataset for high-dimensional actions.
> >         This is because the curse of dimensionality will apply for larger action spaces when optimizing for $\mathcal{D}_{student}$.
> >     1.  **MAIL** has been tested on cases where the number of end-effectors is different from teacher to student, although other forms of morphological differences exist.
> >     1.  The state-visitation distribution of demonstration trajectories must overlap with that of the student agent; this overlap must contain the equilibrium states of the demonstration.
> >         For example, a one-gripper agent cannot reach a demonstration state where two objects are moving simultaneously, but it *can* reach a state where both objects are stable at their goal locations (equilibrium).
> >     1. When we discuss generalizability for the case n<=m, our chosen method to tackle morphological mismatch was to simply use fewer arms on the student robot, in lieu of trajectory optimization.
> >     This is an inefficient approach, since we just don't use some arms of the student robot.
> >     1.  **MAIL** cannot work when the student robot is unable to reach the goal or intermediate states in the demonstration.
> >         For example, in trying to open a flimsy bag with two handles, both end-effectors may simultaneously be needed to keep the bag open.
> >     1.  **MAIL** builds a separate policy for each student robot morphology and each task. While it is possible to train a multi-task policy conditioned on a given task (provided as an embedding or a natural language instruction), extending MAIL to output policies for a variable number of end-effectors would require more careful consideration.
> >         Subsequent work could learn a single policy conditioned on the desired morphology - another way to think about a base model for generalized LfD.
> >
> > 1.  **Incomplete Technical Explanations**: Addressed in response to questions for rebuttal, \#1.
> >
> > 1.  **Limited Exploration of Alternative Optimization Techniques**: Addressed in response to questions for rebuttal, \#4.
> >
> > 1.  **Lack of Discussion on Dataset Variability**:
> >     We have 100 demonstrations provided by the teacher, mentioned on line 193 (sec 3.4).
> >     The diversity of the task comes from the initial conditions for these demonstrations, which are sampled from the task distribution $\boldsymbol{v}_d \sim \mathcal{V}$.
> >     This variability in the initial state adds diversity to the dataset.
> >     The quality and performance of these teacher demonstrations were briefly discussed in the ablations (Sec A.2.4 and Table 1c).
> >
> >
> >     We will also add information about recording demonstrations for the teacher dataset.
> >     Demonstrations come from a scripted policy.
> >     For ClothFold, the teacher has two end-effectors and picks two corners of the cloth to move them towards the other two corners.
> >     For DryCloth, the teacher has two end-effectors and picks two corners of the cloth to move them to the other side of the rack. They maintain the same distance between each other during the move to ensure the cloth is spread out when it hangs on the rack.
> >     For ThreeBoxes, the teacher has three end-effectors. It picks up all the boxes and places them in their respective goals.
> >
> >     We will add more dataset information in a separate section in the appendix.
> >     Please let us know if you would like more information to be added.
> >
> > We hope this response is adequately responding to your questions and concerns about the paper.
> >
> > **References**
> >
> > [1] G. Williams et al., "Information theoretic MPC for model-based reinforcement learning," 2017 IEEE International Conference on Robotics and Automation (ICRA), Singapore, 2017, pp. 1714-1721, doi: 10.1109/ICRA.2017.7989202.

---

> > > ### Comment · Reviewer_nE7X · 2023-08-13
> > > **Response to author**
> > >
> > > Thank you for the in-depth and detailed responses.
> > > The additional technical details and clarifications are useful and are much appreciated.
> > >
> > >
> > > > The learned dynamics model will have some error compared to true dynamics, described as the speed-accuracy tradeoff in section 3.
> > >
> > > > Another effect of using the learned dynamics is that trajectory optimization is a lot faster because the dynamics is about 50x faster (see line 169 or Figure 8).
> > >
> > > Figure 8 (in the supp) is just photos of the visualisation of the cloth at some position. I'm not sure what is the take-away message by just looking at Fig. 8. (e.g. is it to judge if the predicted state from the learned model is similar to the simulator? The high-dimensionality of the cloth makes it a bit hard to do so). Perhaps there would be a more quantitative (and intuitive) way to illustrate your point of "speed-accuracy tradeoff".
> > >
> > > >  [CHOME, STOMP, etc.] We tried training such an inverse dynamics model earlier [...] However, Model Predictive Path Integral (MPPI)is a suitable [...] method to compare against. [...] we will update you as soon as it finishes
> > >
> > > Thank you for the clarification on this aspect; that makes sense. Looking forward to seeing the relevant results.
> > >
> > > ----------------------
> > >
> > > Your other responses (that are not directly addressed above) on the added elaboration/clarifications on limitations, dataset, optimisers are noted and much appreciated.

---

> > > > ### Author Response · Authors · 2023-08-14
> > > > **Response to reviewer nE7X**
> > > >
> > > > Thank you for your kind response to our comments.
> > > >
> > > > > Perhaps there would be a more quantitative (and intuitive) way to illustrate your point of “speed-accuracy tradeoff”.
> > > >
> > > > We agree that we only gave a qualitative account of the accuracy, even though we had quantitative estimates for the speed (162 fps v/s 3.4 fps).
> > > > We wish to improve upon it in this response.
> > > > To quantitatively illustrate the point about accuracy, we have evaluated our learned dynamics model. We provide the simulator and the learned model some initial state $s$ and action $a$.
> > > > These are randomly sampled data from the training distribution.
> > > > We then obtain the new state from the simulator, $s2_{true}$, and the learned model, $s2_{pred}$.
> > > > These states are point clouds representing the cloth.
> > > > We measure the error between $s2_{true}$ and $s2_{pred}$ using the chamfer distance, commonly used to compare point clouds.
> > > > We also compute a metric for how much the cloth truly moved in the simulator, by measuring the chamfer distance between $s$ and $s2_{true}$.
> > > > We ran this experiment for 100 such transitions from $(s, a)$ to obtain $s2_{true}$ and $s2_{pred}$.
> > > > We observed an error of about 0.17 meters between the true and predicted cloth states, for an actual movement of about 0.72 meters between $s$ and $s2_{true}$.
> > > > This error is not as low as one would hope for a learned dynamics model.
> > > > It is because our actions are large deltas that enable a full pick and a place (line 289).
> > > > This is also why other established dynamics models did  not have good results (e.g. GNS).
> > > > However, our learned dynamics model gives us a 50x speed up (line 169).
> > > > This is the speed-accuracy trade-off: we accept an error between true and predicted particle positions, in return for faster trajectory optimization.
> > > >
> > > > > Thank you for the clarification on this aspect; that makes sense. Looking forward to seeing the relevant results.
> > > >
> > > > With regards to MPPI, the results are in table 1 in this pdf.
> > > >
> > > > https://drive.google.com/file/d/1xvTp_U0aWhtRd-8SJ8fElkEuIKMBXyTv/view?usp=sharing
> > > >
> > > > We tried MPPI with 23 different sets of hyperparameters (number of timesteps and samples), and chose the best one. It
> > > > gave reasonable results but the performance was similar to CMA-ES; it was not as performant as CEM. We will include these
> > > > parameters in the appendix in the final version.

---

> > > > > ### Comment · Reviewer_nE7X · 2023-08-14
> > > > > **Responses to added results**
> > > > >
> > > > > Thank you for the detail response to my previous questions.
> > > > >
> > > > > With regards to your comments on speed-accuracy trade-off, that is a fair point and I think that's a fair trade-off.
> > > > >
> > > > > Thanks for the additional comparisons and the results looks good.

---

### Official Review · Reviewer_igC1 · 2023-07-12

**Confidence:** 3
**Originality:** Good
**Technical Quality:** Very Good
**Clarity Of Presentation:** Good
**Impact:** 3

**Recommendation:**

Weak Accept: I recommend accepting the paper, but will not argue for my recommendation if the majority of other reviewers have a different opinion.

**Review:**

The authors propose an intuitive recipe for addressing a challenging and important problem in robotics: cross-morphology imitation learning. Progress in this area can unlock significant advances, such as robot learning from videos of humans as well as reusable expert data across a range of robot morphologies. The proposed method is technically sound and achieves strong improvement over baselines in the experiments. Cloth manipulation is a difficult task, and the video attachment demonstrates the proficiency of the single-picker student in accomplishing the task demonstrated by the dual-picker teacher.

The paper has some weaknesses as well. First, the method is simple: essentially, it simply trains a forward dynamics model then performs the cross-entropy method, which is a well-known formula (e.g., visual foresight [https://arxiv.org/abs/1812.00568]). While the dynamics model is motivated by the use of the expensive cloth simulator, even this component is not required if fast forward simulation is available. Its application to cross-morphology IL is novel as far as I know, but given the large body of work in imitation from observation (IfO) that I am not fully familiar with, I would not be surprised if another reviewer is aware of a similar work that has tried this approach. Second, evaluation is significant but a bit limited. The rearrangement task is very simple as the authors note, and the cloth manipulation involves only two softgym tasks. Also, the real-world evaluation is limited to demonstrating sim2real policy transfer rather than learning from real data. Given that there is no simulator of the real world, it would be interesting if a visual dynamics model learned on real data could be used in the proposed method. Third, the clarity of the paper would benefit from several clarifications and added details, some of which I list below:

- How is the dataset of random actions for training the dynamics model generated? What is the random action policy, the range of the action deltas, the episode horizon, etc? This would be good to add to the appendix as adequate coverage of the state space is essential for the dynamics model to be useful in MAIL.
- How many particle positions N are used in the state representation?
- Section 3.3 says "we use them with the simulator to obtain the optimized trajectories" -- does D_student consist of states from the dynamics model or the simulator?
- How are demonstrations provided? Is it a scripted policy? What are the details of the scripted policy?
- Figure 3 shows mean and error bars across seeds - are these training seeds or evaluation seeds? If the latter, is there only a single training seed per algorithm?
- The sim-to-real gap appears to be extremely large for the Three Boxes task -- how does an image-based policy trained in sim transfer zero-shot to the real world in this case? Is there a state estimation pipeline in real?
- The authors note an emergent behavior of straightening the cloth out on the plank. Is this behavior captured in D_student or generated by the LfD algorithm?
- The authors note GNS performs poorly because it is trained on small action deltas. Why not just train it on the larger action deltas for a more fair comparison?

**Quality Of The Limitations Section:**

Limitations are addressed clearly

**Questions For Rebuttal:**

See weaknesses in the Review section.

**Robotics Focus:**

Sufficient demonstration on hardware

**Summary Of Paper:**

This paper presents MAIL, a new imitation learning algorithm for learning across large differences in teacher and student action spaces. For example, the teacher could demonstrate with two hands while the student robot must use a single manipulator. The key idea in the approach is to perform trajectory optimization in the student's action space such that the distance between the final state in the student trajectory and that of the teacher's trajectory is minimized. To make this process tractable when forward simulation is computationally expensive, the authors propose training a forward dynamics model that can be queried instead. The student dataset created through this process can then be input to an off-the-shelf IL or RLfD (reinforcement learning from demonstration) algorithm that produces a proficient student policy. The authors evaluate their method on cloth manipulation and box rearrangement in simulation and demonstrate transfer to real hardware.

**Summary Of Recommendation:**

**Post-Rebuttal Update**: I am keeping my Weak Accept score. The authors have addressed most of my concerns, but I am still not convinced that the method is particularly novel. I still recommend acceptance as the paper has other strengths.

I recommend acceptance contingent on making the paper clearer in the ways I have noted in the review. This should be very doable in the rebuttal period as I haven't explicitly asked for any additional experiments. However, if the other reviewers disagree, especially if they have found highly related work I am not aware of, I may reconsider.

---

> ### Author Response · Authors · 2023-08-11
> **Response to Reviewer igC1 (1/2)**
>
> We thank you for the insightful comments on our work. We have addressed your concerns and queries in this response.
> Each comment of yours is in bold, followed by our response.
>
> 1. **it simply trains a forward dynamics model then performs the cross-entropy method, which is a well-known formula**:
>     Although each existing method is not new, we have novel applications of these methods. For example, we use trajectory optimization to get trajectories across embodiments with different numbers of end-effectors. Moreover, the extension of a previous method like DMfD to use suboptimal trajectories is a novel contribution.
>     Further, other aspects of the work were non-trivial, such as the selective deployment of a learned dynamics model v/s the simulator.
>
> 1.  **evaluation is significant but a bit limited**: We understand the point made here. We chose the toy task to showcase 3-to-2, 3-to-1, and 2-to-1 end-effector transfer. We chose the ClothFold task, as it is a quintessential cloth manipulation task that is very high-dimensional and challenging. We created the DryCloth task from scratch as a way to get an even harder task -- 3D cloth manipulation where we are concerned with obstacles and collision. Thus, we chose three tasks along a spectrum from easy to hard, although it could be possible to test on more tasks, given enough time.
>
> 1.  **the real-world evaluation is limited to demonstrating sim2real policy transfer rather than learning from real data. Given that there is no simulator of the real world, it would be interesting if a visual dynamics model learned on real data could be used in the proposed method**:
>     We thank you for this idea! It would be interesting to see how this would work with real-world data.
>     We would first need a state estimation pipeline for the objects we work with since **MAIL** needs full state information during demonstration.
>     Alternatively, another line of future work could be to extend **MAIL** to use observations alone for cross-morphological trajectory optimization.
>
> 1.  **How is the dataset of random actions for training the dynamics model generated? What is the random action policy, the range of the action deltas, the episode horizon, etc?**:
>     For ClothFold, our random action policy is to pick a random cloth particle and move the particle to a random goal location within the action space. For DryCloth, the random action policy is to pick a random cloth particle, and move it to a random goal location around the drying rack, to learn cloth interactions around the rack. For completeness, we also trained a forward dynamics model for the ThreeBoxes task. Here, the random action policy is to pick the boxes in order and sample a random place location within the action space.
>     Our episode horizon is three. Our actions are pick-and-place actions, and the action delta is in the full range of visibility of the camera. For DryCloth, this limit is $(-0.5, 0., -0.5)$ to $(0.5, 0.7, 0.5)$. For ClothFold it is $(-0.9, 0., -0.9)$ to $(0.9, 0.7, 0.9)$. For ThreeBoxes it is $-0.1$ to $1.35$.
>     We will add this to the appendix in the final version of the paper.
> 1.  **How many particle positions N are used in the state representation?**: Our cloth is discretized into an 80x80 grid of particles, giving a total of 6400 particles. We will add this to the appendix description of tasks.
> 1.  **Section 3.3 says, ``we use them with the simulator to obtain the optimized trajectories" -- does D\_student consist of states from the dynamics model or the simulator?**: The final $\mathcal{D}_{student}$ has states from the simulator after optimized actions have been obtained via CEM. We will clarify this in Sec. 3.3 (Indirect trajectory optimization).
> 1.  **How are demonstrations provided? Is it a scripted policy? What are the details of the scripted policy?**: Demonstrations come from a scripted policy.
>     For ClothFold, the teacher has two end-effectors and picks two corners of the cloth to move them towards the other two corners.
>     For DryCloth, the teacher has two end-effectors and picks two corners of the cloth to move them to the other side of the rack. They maintain the same distance between each other during the move to ensure the cloth is spread out when it hangs on the rack.
>     For ThreeBoxes, the teacher has three end-effectors. It picks up all the boxes and places them in their respective goals.
>     We will clarify this scripted policy in the appendix.
> 1.  **Figure 3 shows mean and error bars across seeds - are these training seeds or evaluation seeds? If the latter, is there only a single training seed per algorithm?**:
>     The mean and error bars, and all data in SOTA comparisons, are across 5 evaluation seeds. These are different from the training seeds, which are another set of 5 seeds. Each model from a single training seed is evaluated across 5 evaluation seeds, and so these results are a combination of 25 different evaluations.
>
> (... continued in comment 2/2)

---

> > ### Author Response · Authors · 2023-08-11
> > **Response to Reviewer igC1 (2/2)**
> >
> > (... continued from comment 1/2)
> >
> > 9.  **The sim-to-real gap appears to be extremely large for the Three Boxes task -- how does an image-based policy trained in sim transfer zero-shot to the real world in this case? Is there a state estimation pipeline in real?**
> >     It is true that there isn't a perfect correspondence between sim and real for ThreeBoxes.
> >     To reduce the sim-to-real gap, we have a pipeline to segment images and determine the foreground (box) from the background (table).
> >     We use this to pre-process the image going into the policy.
> >     We do not have a state-estimation pipeline in real; we could get better results with such a pipeline, but then our policy (and real-world setup) would be state-dependent.
> > 1.  **The authors note an emergent behavior of straightening the cloth out on the plank. Is this behavior captured in D\_student or generated by the LfD algorithm?**: This behavior is seen first in $\mathcal{D}_{student}$, and is then improved upon while training the LfD algorithm.
> > 1.  **The authors note GNS performs poorly because it is trained on small action deltas. Why not just train it on the larger action deltas for a more fair comparison?**:
> >     Our initial experiments were done using the larger action deltas, as that is what we have for our pick-and-place actions.
> >     However, we were getting poor training results, and the dynamics were very off. Thus, we switched to small action deltas, following the implementation from VCD [1].
> >
> > We hope this response is adequately responding to your questions and concerns about the paper.
> >
> > **References**
> >
> > [1] X. Lin et al., "Learning Visible Connectivity Dynamics for Cloth Smoothing," 2021 Conference on Robot Learning.

---

> > > ### Comment · Reviewer_igC1 · 2023-08-13
> > > **Thanks**
> > >
> > > Thanks to authors for addressing my concerns. Confirming I have read the responses and will consider them in the next phase. No further clarifications needed at this time.

---

### Official Review · Reviewer_ExJj · 2023-07-18

**Confidence:** 4
**Originality:** Fair
**Technical Quality:** Very Good
**Clarity Of Presentation:** Very Good
**Impact:** 3

**Recommendation:**

Weak Accept: I recommend accepting the paper, but will not argue for my recommendation if the majority of other reviewers have a different opinion.

**Review:**

Quality/ Clarity
--------------------
Paper is overall well written and presented. However, some sections are too reliant on appendices and cited work. For example, a short description of DMfD would help to make the paper more self-contained. Furthermore, the abalation study is not really useful without the results. Either incorporate results into the section or move entirely into appendix.

Originality
---------------
The proposed method is mainly a combination of existing methods. Utilising full state information for task transfer and then training the policy on images is an interesting idea but may be restrictive for certain problem settings.

Significance
-----------------
The proposed framework is shown to work well and is mostly practical enough to be utilised. Being able to demonstrate a task naturally and transfer to a robot with a significantly different embodyment has real use-case scenarios.

Strengths
----------------
- Method works well and generalises to varying inital conditions
- Thorough experiments and shown to transfer to real world hardware
- Video is nicely put together and helps in understanding the paper

Weaknesses
-------------------
- Question around novelity since this is mainly a trivial combination of existing work
- Applicability to certain problem settings may be limited due to requiring full state information
- Method explanations are lacking and could be more self contained rather than simply citing work being used
- Some sections rely too much on content in appendix



**Quality Of The Limitations Section:**

Limitations are addressed clearly

**Questions For Rebuttal:**

For the 4.3 experiment I believe you are training separate policies for each student morphology as opposed to one policy which generalises to different morphologies. If so, can you please make this clearer in the text?

**Robotics Focus:**

Sufficient demonstration on hardware

**Summary Of Paper:**

Authors propose a novel learning from demonstration (LfD) framework MAIL which allows for teacher and student morphologies to vary greatly. The particular discrepency they focus on are when the number of end effectors differ. Their method consists of learning a dynamics model which is used by a trajectory optimiser to convert demonstrations into the student's morphology. Finally, this trajectory is used by an existing LfD method which learns a policy in state and image space, intended to generalise to arbitrary initial environment states. They demonstrate their method in a wide range of experimental scenarios, both in simulation and hardware, and show their method outperforms baseline methods and is able to generalise to different morphologies.

**Summary Of Recommendation:**

The main strength of this paper is the experimental section. However, overall the paper lacks originality combined with vague method descriptions.

---

> ### Author Response · Authors · 2023-08-11
> **Response to Reviewer ExJj**
>
> We thank you for the insightful comments on our work. We have addressed your concerns and queries in this response.
> Each comment of yours is in bold, followed by our response.
>
>
> 1.  **Some sections rely too much on content in appendix**: We understand that the ablations are primarily located in the appendix. Thus, we will remove the ablation section 4.4, and instead refer the reader to the appendix for all ablations we have performed.
>
> 1.  **Question around novelity since this is mainly a trivial combination of existing work**: Although each existing method is not new, we have novel applications of these methods. For example, we use trajectory optimization to get trajectories across embodiments with different numbers of end-effectors. Moreover, the extension of a previous method like DMfD to use suboptimal trajectories is a novel contribution. Further, other aspects of the combination were non-trivial, such as the selective deployment of a learned dynamics model v/s the simulator.
>
> 1.  **Applicability to certain problem settings may be limited due to requiring full-state information**: Yes, that is true, and we have mentioned that full-state information for demonstrations and simulation training. It is a current limitation described in Sec. 4.5 (Limitations).
> 1.  **Method explanations are lacking and could be more self-contained rather than simply citing work being used** :
>     We will add more technical explanations to our paper.
>     Specifically, the final version will include details of DMfD instead of simply citing it.
>     We will add details on how our trajectory optimization is formulated, some of which are already present: the fitness function, dynamics function, CEM hyperparameters, action parameters chosen to be optimized, and obtaining states once the actions are optimized.
>
>     Similarly, we will include these details about the adaptation of LfD methods: the optimized demonstrations $\mathcal{D}_{student}$ we provide, details about the optimized dataset and its characteristics, the chosen LfD method, and some other alternatives (this can include offline alternatives that don't need environment interaction, so we don't need the simulator or the learned dynamics model for them).
>
>     If there is any other relevant detail you think we may have missed, please let us know.
>
> 1.  **For the 4.3 experiment, I believe you are training separate policies for each student morphology as opposed to one policy that generalizes to different morphologies. If so, can you please make this clearer in the text?** Yes, that is correct. We will make it clearer in Sec. 4.3, and link to the limitations section where this is also mentioned.
>
>
> We hope this response is adequately responding to your questions and concerns about the paper.

---

### Author Response · Authors · 2023-08-11
**Response from Authors**

We thank the reviewers for their valuable and insightful comments on our work. We particularly appreciate the thoughtful, encouraging nature of the reviews and accompanying suggestions for improvement. We address them comprehensively in this discussion phase.

---

### Decision · Program_Chairs · 2023-08-30

**Decision:**

Accept (Poster)

**Comment:**

This paper introduced a cross-morphology imitation learning method, which aims to learn from demonstrations where the teacher robot's morphology is significantly different from the student's, e.g., different numbers of end-effectors. The proposed approach leveraged a learned forward dynamics model to perform the cross-entropy method such that the distance between the final state in the student's trajectory and that of the teacher's trajectory is minimized. Evaluations were conducted on cloth manipulation and box rearrangement tasks in simulation, and sim2real results have been shown on a physical robot arm.

This paper received positive ratings from four reviewers, who gave Weak Accept after the rebuttal. However, there was a lack of enthusiastic support. Some lingering issues were brought up by the reviewers, including the limited novelty of the proposed method and the experiments on toy problems, which are also acknowledged by the authors to some extent. Toward the end of the discussion period, Reviewer ExJj expressed that "Inclined to reject this paper since authors have been unable to convince me that their approach isn't just a trivial combination of existing methods." Reviewer igC1 "will keep it as a weak accept as I'm not convinced the method is very novel."

The AC read the paper after going through the reviews, rebuttal, and post-rebuttal discussions in detail. While there is ample room to improve this work, it has offered a reasonable solution to a well-motivated problem. To encourage this line of work, the AC would recommend this paper to be accepted, but with a low confidence, and leave it to the PCs for the final decision.